# Technical Note: The Role of Evolving Surface Tension in the Formation of Cloud Droplets

James F. Davies[1], Andreas Zuend[2], Kevin R. Wilson[3]

[1]Department of Chemistry, University of California Riverside, CA USA
[2]Department of Atmospheric and Oceanic Sciences, McGill University, Montreal, Quebec, Canada
[3]Chemical Sciences Division, Lawrence Berkeley National Laboratory, Berkeley, CA USA

*Correspondence to*: James F. Davies (jfdavies@ucr.edu)

**Abstract.** The role of surface tension ($\sigma$) in cloud droplet activation has long been ambiguous. Recent studies have reported observations attributed to the effects of an evolving surface tension in the activation
process. However, adoption of a surface-mediated activation mechanism has been slow and many studies continue to neglect the composition-dependence of aerosol/droplet surface tension, using instead a value equal to the surface tension of pure water ($\sigma_w$). In this technical note, we clearly describe the fundamental role of surface tension in the activation of multicomponent aerosol particles into cloud droplets. It is demonstrated that the effects of surface tension in the activation process depend primarily on the evolution
of surface tension with droplet size, typically varying in the range $0.5\sigma_w \lesssim \sigma \leq \sigma_w$ due to the partitioning of organic species with a high surface affinity. We go on to report some recent laboratory observations that exhibit behavior that may be associated with surface tension effects, and propose a measurement coordinate that will allow surface tension effects to be better identified using standard atmospheric measurement techniques. Unfortunately, interpreting observations using theory based on surface film and liquid-liquid
phase separation models remains a challenge. Our findings highlight the need for experimental measurements that better reveal the role of composition-dependent surface tensions, critical for advancing predictive theories and parameterizations of cloud droplet activation.

## 1 Introduction

The formation of a cloud involves a complex series of steps as nanometer sized aerosol particles, termed cloud condensation nuclei (CCN), grow by condensation of water vapor to become supermicron-sized cloud droplets, in a process known as CCN activation. Activation depends on physicochemical properties of the aerosol, such as hygroscopicity and surface tension, as well as atmospheric conditions, such as temperature and humidity. To accurately predict cloud formation and properties, these factors must be included in modelling schemes. However, due to computational limitations, approximations and simplifications are needed, which often obscures the underlying physics and may limit the accuracy of predictions. A key challenge is in the development of a simple model that captures the basic processes involved in CCN activation, while allowing complicating factors such as surface tension variability, solubility and phase separation to be included in a physically representative manner. In this note, we focus on the role of surface tension, and discuss the limitations of current approximations in light of recently published works that reveal how it is primarily the evolution of surface tension that impacts the activation process.

In recent publications, the role of surface tension in the activation of aerosol particles to cloud droplets has been reexamined (Forestieri et al., 2018; Ovadnevaite et al., 2017; Ruehl et al., 2016) . These studies show that the evolution of surface tension can have a large effect during the activation process compared to when surface tension is assumed to be a static parameter. It is well established that surface tension is a factor in activation, and that dissolved species can suppress surface tension (Li et al., 1998). Traditionally, however, surface tension has been reduced to a fixed term in the Köhler equation (Abdul-Razzak and Ghan, 2000; Facchini et al., 2000; Petters and Kreidenweis, 2007) and is usually given a temperature-independent value equal to that of pure water at 25 °C. This is because for any decrease in surface tension due to bulk–surface partitioning and surface adsorption, it is assumed that there is an increase in the solution water activity because adsorbed material, previously acting as a hygroscopic solute, is removed from the droplet (bulk) solution (Fuentes et al., 2011; Prisle et al., 2008; Sorjamaa et al., 2004). Thus, the effects approximately cancel out in the calculation of a droplet's equilibrium saturation ratio via the Köhler equation and so are

often neglected. Furthermore, it has been shown in some cases that there is insufficient material in a droplet at the sizes approaching activation to sustain a surface tension depression (Asa-Awuku et al., 2009; Prisle et al., 2010). The lack of experimental evidence to the contrary has led to the adoption of these assumptions in popular single-parameter models, such as $\kappa$-Köhler theory, that reduce the complexity of the activation

process (Petters and Kreidenweis, 2007). These parameterizations provide a compact and useful means of relating key observables, such as the critical supersaturation and activation diameter, to hygroscopicity and allow a general comparison between systems with arbitrary compositions. The $\kappa$-Köhler framework has also been adapted to account for surface tension effects (Petters and Kreidenweis, 2013). However, a single parameter implementation cannot account for the full effects of an evolving surface tension and, by omitting

the microphysical processes associated with bulk–surface partitioning, the presence and magnitudes of any surface effects are often difficult to ascertain.

In the works of Ruehl and coworkers (Ruehl et al., 2016; Ruehl and Wilson, 2014), Forestieri and coworkers (Forestieri et al., 2018) and Ovadnevaite and coworkers (Ovadnevaite et al., 2017), laboratory and observation-based measurements, respectively, combined with a partitioning model have revealed key

signatures of surface tension lowering in the activation process due to non-surfactant organic compounds. Notably, that a modification of the Köhler curve can result in lower critical supersaturations and vastly different droplet sizes at activation compared to the expectation when assuming a constant surface tension. Dynamic factors may also play a role, as discussed by Nozière and coworkers (Nozière et al., 2014) who have shown that surface tension can vary over time due to slow changes in the bulk-surface partitioning of

material, leading to a time-dependence in the role of surface tension. This may be especially important for droplets that initially contain micelles or oligomers that exhibit slow breakdown kinetics and diffusion. Furthermore, surface partitioning may be influenced by non-surface active components in the system, such as the presence of inorganic material and co-solutes (Asa-Awuku et al., 2008; Boyer et al., 2016; Boyer and Dutcher, 2017; Frosch et al., 2011; Petters and Petters, 2016; Prisle et al., 2011; Svenningsson et al., 2006;

Wang et al., 2014). Other factors that have been shown to influence the shape of Köhler curves are: (1)

solute dissolution, encompassing both water-solubility and solubility kinetics (Asa-Awuku and Nenes, 2007; Bilde and Svenningsson, 2017; McFiggans et al., 2006; Petters and Kreidenweis, 2008; Shulman et al., 1996), (2) liquid–liquid phase separation (i.e. limited liquid–liquid solubility) (Rastak et al., 2017; Renbaum-Wolff et al., 2016), and (3) the dynamic condensation (or gas–particle partitioning) of organic

vapors (Topping et al., 2013; Topping and McFiggans, 2012) (Topping et al., 2013; Topping and McFiggans, 2012). While measured cloud droplet number concentrations in the atmosphere have been explained in several cases with simple parameterizations that neglect dynamic surface effects (Nguyen et al., 2017; Petters et al., 2016), there are many observations that are not fully explained in such simple terms and in those cases a substantial population of CCN may exhibit behavior characteristic of surface effects

(Collins et al., 2016; Good et al., 2010; Ovadnevaite et al., 2011; Yakobi-Hancock et al., 2014). In order to gain a robust and predictive understanding of CCN activation, a molecular-level theory must be developed and adopted by the atmospheric chemistry community.

In this technical note, we offer a perspective on the role of surface tension in the activation process, drawing on recent studies and interpretations of cloud droplet activation measurements (e.g. (Forestieri et al., 2018;

Ovadnevaite et al., 2017; Ruehl et al., 2016; Ruehl and Wilson, 2014). We go on to discuss how surface tension may be considered in the activation process, and finally present some new data highlighting potential indicators of surface tension effects in measurements of critical supersaturation. Our aim is to provide a platform for discussion and help foster a molecular-based interpretation of the role of organic material in the activation of aerosol to cloud droplets.

**2 Clarifying how Surface Tension Alters Cloud Droplet Activation**

On a fundamental level, the influence of surface tension on droplet activation is straightforward and was discussed in the late 1990's for aerosol particles containing surfactants (Li et al., 1998). Unfortunately, the simplicity of the role of surface tension in CCN activation has been lost in the complex descriptions of surface and phase-partitioning models, limiting the broader application of the insights gained from recent

experimental results. For context, we begin our discussion with Köhler theory (Köhler, 1936), which

describes the thermodynamic conditions required for CCN activation based on two contributions that control the equilibrium (saturation) vapor pressure of water above a liquid surface. The classic Köhler equation is often written as (Petters and Kreidenweis, 2007):

$$S_d = \frac{p_{w,d}(D)}{p_w^0} = a_w \exp\left[\frac{4\,M_w\,\sigma}{RT\rho_w D}\right], \tag{1}$$

where $S_d$ is the equilibrium saturation ratio of water in the vapor phase surrounding a droplet surface, $p_{w,d}$ (D) is the equilibrium partial pressure of water vapor above a droplet (subscript $d$) of diameter $D$ and a certain chemical composition, $p_w^0$ is the pressure of water above a flat, macroscopic surface of pure liquid water at temperature $T$, $a_w$ is the mole-fraction-based water activity of the droplet solution, $M_w$ is the molecular mass of water, $\sigma$ is the surface tension of the particle (at air-liquid interface), $R$ is the ideal gas

constant, and $\rho_w$ is the density of liquid water at $T$. The water activity contribution, known as the solute or Raoult effect, describes a lowering of the equilibrium water vapor pressure above a liquid surface due to the presence of dissolved (hygroscopic) species that reduce the water activity to a value below 1. The second contribution, known as the Kelvin effect, describes an increase in the equilibrium water vapor pressure above a microscopic curved surface and is dependent upon the surface area-to-volume ratio of the droplet

(a size effect) and the surface tension, i.e. the gas-liquid interfacial energy per unit area of surface. The latter term arises from the energy associated with creating and maintaining a certain surface area and, thus, is reduced when the surface tension is lowered or when the droplet size increases, leading to a smaller surface-to-volume ratio. The magnitude of the Kelvin effect scales with the inverse of the droplet radius and is sometimes referred to as the "curvature effect". The combined contributions from the Raoult and

Kelvin effects in Köhler theory define a thermodynamic barrier to droplet growth. It is important to note that the Köhler equation describes the specific saturation ratio $S_d$ in thermodynamic equilibrium with a certain solution droplet of interest; however, the value of $S_d$ may differ from that of the environmental saturation ratio, $S_{env}$, present in the air parcel containing the droplet, since $S_{env}$ is established by an interplay of moist thermodynamic processes. The environmental saturation ratio is defined by $S_{env} = \frac{p_w}{p_w^0}$,

where $p_w$ is the partial pressure of water in air at a specific location and time, irrespective of the presence

or absence of aerosols and cloud droplets. The global maximum in a Köhler curve marks the point of

activation for a certain CCN, as shown in Figure 1A. For conditions of $S_d \leq S_{\text{env}}$, e.g. in a rising,

adiabatically expanding air parcel, aqueous CCN equilibrate relatively quickly to their environmental

conditions such that $S_d = S_{\text{env}}$ is maintained (stable growth/evaporation). However, when $S_{\text{env}}$ exceeds

the global maximum in $S_d$ for a certain CCN, such an equilibration becomes unattainable and net

condensation of water prevails, leading to so-called unstable condensational growth for as long as $S_{\text{env}} >$

$S_d$ holds (while $S_d$ varies according to the pertaining Köhler curve).

A Köhler curve shows the relationship between the droplet's equilibrium water vapor saturation ratio and

the wet droplet size. The wet droplet size, or more specifically the chemical composition (solute

concentration), solubility and non-ideal mixing determines the water activity, while the size and surface

tension determine the Kelvin effect. Since the solute concentration changes with the droplet size, e.g. during

net growth conditions when the environmental saturation ratio in an air parcel increases and water vapor

condenses, the water activity term varies accordingly, typically in a non-linear manner. The Kelvin term

should also change with droplet size, both due to the changing surface-to-volume ratio of the droplet and

changes in surface tension as a result of changes in solute concentration and related surface composition.

In most scenarios, only the diameter change is accounted for while the surface tension is assumed to remain

constant, usually with the value for pure water ($\sigma = \sigma_w \approx 72$ mN m$^{-1}$ at 298 K). This oversimplifies the

problem, especially when organic solutes are present that adsorb at the surface of the growing droplet. In

experimental studies where only the critical supersaturation or critical dry diameter are measured, this

assumption can lead to errors, as the complexity of the system may not allow for such simplified treatments

to yield sufficiently accurate representations of a real-world problem. In such cases, one must consider how

the changing size of the droplet (or, again, more specifically the solution composition) results in changes

in bulk-surface partitioning and ultimately surface tension.

Recent work has shown that a rigorous account of bulk–surface partitioning leads to complex Köhler curves

whose shapes are often difficult to interpret (Ruehl et al., 2016). These shapes can be more easily understood

by considering a very simple example, shown in Figure 1B. Here, a series of fixed surface tension (*iso-σ*)

Köhler curves with values ranging from 72 mN m$^{-1}$ to 30 mN m$^{-1}$ are shown, using the same 50 nm particle

(of ammonium sulfate) from Figure 1A. Distinct schematic dependences of the surface tension on the

droplet size, representative of the types of dependence that might be encountered in real aerosol, are

imposed for the purpose of illustration, shown in the lower panel of Figure 1B. At each point along these

dependences, the saturation ratio will be determined by the position on the Köhler curve corresponding to

that specific surface tension. This means that instead of following a single trajectory along an *iso-σ* curve,

the system is better envisioned as traversing across these curves, producing a very different final shape to

the Köhler curve than would normally be expected. In the case of a linear dependence of $\sigma$ on $D$ (black

curve), the Köhler curve cuts across the *iso-σ* lines until $\sigma$ reaches the value of pure water. In this case, this

coincides with reaching the maximum in the droplet's saturation ratio and thus reflects the activation point.

In the case shown in red, activation occurs prior to the surface tension returning to the value of pure water.

In the case shown in green, activation follows a pseudo two-step process, where initially a large increase in

size occurs for a small increase in supersaturation, and thus the droplet may appear to be activated, while

in this case true activation occurs at the point corresponding to the intersection with the Köhler curve of $\sigma$

$= \sigma_w$. It is important to note that it is not necessarily the magnitude of the suppression of surface tension

that drives unusual activation behavior, but the dependence of the surface tension on concentration, which

is dictated by the droplet size and the thermodynamics of the system.

It is apparent from these examples that an evolving surface tension may introduce abrupt changes in the

activation curve that arise due to the phase behavior of monolayer systems. The most obvious change is

where the surface tension returns to the value of pure water. Following this point, the system will follow

the *iso-σ* curve corresponding to pure water surface tension (under continued and sufficient supersaturation

conditions). Depending on the exact relationship of surface tension and droplet size, this point can mark the

activation barrier of the system, as is the case in Figure 1B for the black and green curves. This also
seemingly supports previous assertions, as discussed in Section 1, that surface tension does not impact
activation, since it is generally argued that at the point of activation the droplet is sufficiently dilute and
essentially exhibits a surface tension like pure water (Fuentes et al., 2011; Prisle et al., 2008; Sorjamaa et
al., 2004). While that is the case in this example, the trajectory of the activation process is significantly
altered by the surface tension history of the droplet. In other words, for the systems described by the black
or green curves in Fig. 1B, the surface tension at the point of CCN activation is that of pure water, yet if
one would assume the surface tension of the droplet to be constantly that of pure water at any size prior to
activation, the critical supersaturation, $SS_{crit}$, would be significantly higher and the critical (wet) activation
diameter, $D_{crit}$, substantially smaller (see the Köhler curve for $\sigma = \sigma_w$). Clearly, an evolving surface tension
prior to activation can matter in such systems and, consequently, knowing the surface tension only at the
point of activation is in a general case insufficient for determining the critical properties at activation (absent
any droplet size measurement), because the position of the maximum in the Köhler curve and, thus, the
droplet size at activation, will depend on the trajectory of surface tension evolution. Moreover, knowing
the surface tension at the activation point only, may not allow for an accurate prediction of whether a droplet
of given dry diameter will activate at a given environmental supersaturation (compare the green curve with
the *iso-σ* curve of 72 mN m$^{-1}$ in Fig. 1B, both having the same surface tension at their points of CCN
activation, yet different critical supersaturations and wet diameters).. Similar conclusions have been drawn
previously (e.g. (Prisle et al., 2008)), although the results of Ruehl et al. (Ruehl et al., 2016) were the first
to verify this experimentally.

**3 Surface Tension Evolution During Activation**

A major consequence of an evolving surface tension is that the droplet size at activation is larger, and the
actual critical supersaturation depends on how surface tension varies in the droplet as it grows. These effects
were directly measured using a thermal gradient chamber (TGC) (Roberts and Nenes, 2010; Ruehl et al.,
2016). In the 2016 work, mixed ammonium sulfate and organic aerosols were introduced into the TGC at

various supersaturations (up to and including the activation supersaturation). The size of the aerosol particles was measured under these conditions, allowing a direct measurement of the stable equilibrium branch of the Köhler curve (i.e. the size up to the point of activation). It was shown that for highly soluble and surface inactive solutes such as ammonium sulfate and sucrose, the data exhibit the behavior expected when assuming an *iso*-σ Köhler curve. However, for the case when organic acids, spanning a range of water-solubilities, were coated on ammonium sulfate particles of specific dry sizes, the measurements clearly show modifications to the Köhler curve in comparison to an *iso*-σ Köhler curve, which were explained by changes in surface tension corresponding to bulk-surface partitioning as predicted by a compressed film model. Without these observations, assuming $\sigma = \sigma_w$, one could attribute the critical supersaturation to a single, apparent $\kappa$ value that describes the hygroscopic effect of the mixture of solutes in the absence of any surface tension constraints (Petters and Kreidenweis, 2007). Low solubility and low hygroscopicity species should exhibit very small $\kappa$ values. However, in the case of suberic acid, for example, the CCN activation data of Ruehl et al. would require a $\kappa$ value of approximately 0.5 for the organic component, assuming a fixed surface tension equal to that of water. This $\kappa$ value is unphysically large considering that the molar volume ratio suggests a value of ~0.13 for suberic acid when assuming full solubility, the surface tension of pure water and ideal mixing with water. A different prediction, accounting for limited solubility and assuming a constant surface tension using the value of the saturated aqueous solution, estimates its value as $\kappa \approx 0.003$ (Kuwata et al., 2013). Moreover, the apparent $\kappa$ value derived from CCN activation data of pure suberic acid particles indicates a value of ~0.001 (Kuwata et al., 2013). Invoking a surface tension model here is the only way to make physical sense of these observations, allowing even low solubility and non-hygroscopic solutes to contribute significantly to the activation efficiency in mixed droplets. The modelling approaches of Ruehl and coworkers (Ruehl et al., 2016) and Ovadnevaite and coworkers (Ovadnevaite et al., 2017) use Köhler theory with either a bulk–surface partitioning model (compressed film model) or an equilibrium gas–particle partitioning and liquid–liquid phase separation (LLPS) model with variable surface tension. Both studies also employed simplified

organic film models, in which the assumption is made that all organic material resides in a water-free surface film, as options for comparison with the more sophisticated approaches.

A bulk–surface partitioning model is comprised of two components: a two-dimensional equation of state that relates the surface tension to the surface concentration, and a corresponding isotherm that relates the surface and bulk solution concentrations. In the work of Ruehl et al., the compressed film (Jura and Harkins, 1946) and Szyszkowski-Langmuir equations of states were compared. The latter has been used in several studies exploring bulk-surface partitioning in organic aerosol (Prisle et al., 2010; Sorjamaa et al., 2004; Topping et al., 2007). The compressed film model reproduced the experimental observations, capturing the complex shapes of the measured Köhler curves. The Szyszkowski-Langmuir method was unable to explain several of the observations, attributed partly to the lack of a two-dimensional phase transition between a film state and a non-film state, which is a unique feature of the compressed film model. Earlier work by Ruehl and coworkers successfully used a van der Waals equation of state to model the behavior of organic and inorganic mixed droplets at high relative humidity (Ruehl and Wilson, 2014), demonstrating the effect of surface tension following the onset of film formation once sufficient organic material was present. While we know the factors that contribute to the equation of state and isotherm for well controlled systems the enormous complexity of atmospheric aerosol presents a significant challenge in developing and utilizing a predictive general model or theoretical framework.

The equilibrium gas–particle partitioning and liquid–liquid phase separation model is based on the Aerosol Inorganic-Organic Mixtures Functional groups Activity Coefficients (AIOMFAC) model (Zuend et al., 2008; 2011), coupled to a relatively simple, phase composition- and morphology-specific surface tension model. A detailed description of this AIOMFAC-based model, its variants and sensitivities to model parameters and assumptions is given in the supplementary information of Ovadnevaite et al (2017). Briefly, the equilibrium gas–particle and liquid–liquid partitioning model is used to predict the phases and their compositions for a bulk mixture "particle", here of known dry composition, at given RH and temperature. For other applications with given total gas + particle input concentrations, the equilibrium condensed-phase

concentrations can be computed as a function of RH (accounting for partitioning of semivolatiles). Assuming spherical particles of a certain dry diameter with a core-shell morphology of liquid phases in the case of LLPS, density information from all constituents is used to compute volume contributions and the size of the particle at elevated RH. In addition, the surface tension of each individual liquid phase is computed as a volume-fraction-weighted mean of the pure-component surface tension values. In the case of LLPS, the surface coverage of the (organic-rich) shell phase is evaluated by considering that it must be greater than or equal to a minimum film thickness (monolayer as a lower limit); this determines whether complete or partial surface coverage applies for a certain wet diameter. The effective surface tension of the whole particle is then computed as the surface-area-weighted mean of the surface tensions of contributing phases. This way the surface tension evolves in a physically reasonable manner as a droplet grows, including the possibility for abrupt transitions from a low surface tension, established due to full organic droplet coverage under LLPS, to partial organic film coverage after monolayer film break-up, and further to complete dissolution of organics in the aqueous inorganic-rich phase (single aqueous phase). This equilibrium model is here referred to as AIOMFAC-EQUIL. We also employed two AIOMFAC-CLLPS model variants, in which the organic constituents are assumed to reside constantly in a separate phase from ammonium sulfate (complete LLPS, an organic film), either with or without water present in that phase, discussed in Section 4. This approach, introduced by Ovadnevaite and coworkers (Ovadnevaite et al., 2017), shows promise due to its ability to predict the existence of a surface tension activation effect consistent with CCN observations taken in marine air containing a nascent ultrafine aerosol size mode. In that study, enhanced CCN activity of ultrafine particles was observed for aerosols consisting of organic material mixed with inorganic salts and acids in North Atlantic marine air masses, which could not be explained when accounting for hygroscopicity or solubility alone (when assuming surface tension of water). While the LLPS-based model and the compressed film model of Ruehl et al. employ different principles and descriptions to account for the surface composition, both agree that gradual surface tension changes dominate the CCN activation process for these systems. Interestingly, while the observations of Ruehl et al. showed activation occurring when the surface tension returns to its maximum value (i.e. that of pure

water), the phase separation model predicted activation prior to the surface tension returning to its

maximum (for ultrafine particles). As discussed by Ovadnevaite et al. (2017), this difference in the surface

tension value reached at the point of CCN activation depends in some cases on the size range of the (dry)

particles considered (for the same dry composition). Ovadnevaite et al. show that for particles of larger dry

diameters (e.g. 175 nm for the case of their aerosol model system), CCN activation is predicted to occur at

a point where the particle's surface tension has reached the value of pure water (see the Supplementary

Information of that study). Hence, both observations are consistent with the picture developed in Figure 1B

and the details depend on the particle size range and functional form describing the change in surface

tension of the system considered. Moreover, it is important to recognize that – regardless of whether the

surface tension is equivalent to or lower than that of pure water at the CCN activation point – a CCN

exhibiting an evolving, lowered surface tension while approaching the activation point during hygroscopic

growth will activate at a lower supersaturation than a CCN of constant surface tension equivalent to the

pure-water value, since the former activates at a larger diameter, as is evident from the examples of the red

and green Kohler curves shown in Fig. 1B. This predicted size effect indicates that it is not generally valid

to assume that all activating CCN will have a surface tension equivalent to or close to that of pure water –

nor is it appropriate to use a single measurement of the surface tension of a multicomponent CCN of known

dry composition at its activation size (only) to determine its Köhler curve, as also noted in previous studies

(Prisle et al., 2010; 2008; Sorjamaa et al., 2004). Furthermore, these model predictions also suggest that

measurements of the surface tension of larger CCN particles (e.g. > 150 nm dry diameter) may not allow

for conclusions about the surface tension of much smaller CCN, e.g. of 50 nm dry diameter.

**4 Identifying Surface Tension Effects from Critical Supersaturation**

Although the LLPS-based and compressed film model of CCN activation have had some success in

capturing surface tension effects, there remain substantial challenges in developing a generalized theory.

One challenge is that CCN techniques do not measure surface tension directly but instead observe the effects

of changes in surface tension, which may often be attributed to other factors. In order to identify surface

tension effects using these techniques, experiments must be performed to maximize the scope of surface

tension effects while minimizing changes in other variables that might influence observations. Here, we

propose new experiments, using a model system as an example, which allows surface tension effects to be

identified in the absence of other complicating factors.

Using a Cloud Condensation Nucleus Counter (CCNC, Droplet Measurement Technology), we measured

the supersaturation required to activate mixed suberic acid and ammonium sulfate particles. Suberic acid

was chosen to represent low solubility oxygenated organic material typical of atmospheric secondary

organic aerosol. It is not a traditional surfactant, but its role in suppressing surface tension and modifying

the shape of the Kohler curve has been previously identified (Ruehl et al., 2016). Ammonium sulfate

particles were generated using an atomizer and dried using silica gel and a Nafion drier with dry $N_2$ counter

flow. The size distribution was measured and a size-selected seed was introduced into a flow tube

containing suberic acid and housed within a furnace oven. The temperature was set to volatilize the organic

material and allow it to condense onto the seed particles up to a desired thickness. The coated particles were

size-selected again and introduced into a particle counter and a Cloud Condensation Nucleus Counter

(CCNC; Droplet Measurement Technologies) at a concentration of around 2000 cm$^{-3}$. The activated fraction

was measured as a function of saturation ratio and the critical supersaturation was determined from the half-

rise times of a sigmoid fit to the data. A range of dry particle sizes and organic volume fractions ($f_{org}$) were

selected for measurements under humidified conditions at room temperature (~ 20 ℃). Notably, the coated

particle (i.e. total size) of the aerosol in its dry state was kept constant across a dataset spanning a range of

organic volume fractions. The CCNC was calibrated with ammonium sulfate particles at room temperature

at regular intervals, although typically the calibration remained stable during continued usage.

The organic volume fraction ($f_{org}$) was varied while maintaining fixed dry particle sizes (Prisle et al., 2010;

Wittbom et al., 2018), ensuring that any surface tension effect would not be masked by changes in the

overall size, in contrast to other studies that allow both coated particle size and organic volume fraction to

vary simultaneously (eg. (Hings et al., 2008; Nguyen et al., 2017)). Figure 2A shows the critical

supersaturation as a function of $f_{org}$ for 100 nm (dry diameter) mixed ammonium sulfate and suberic acid

particles. Remarkably, despite the much lower hygroscopicity of suberic acid relative to ammonium sulfate,

there is very little increase in the required supersaturation as the organic volume fraction increases (while

the ammonium sulfate volume fraction is reduced accordingly). In fact, considering a fixed surface tension

of pure water, a $\kappa_{org}$ value of 0.35 for the organic fraction is required to explain these data (using $\kappa = 0.62$

for ammonium sulfate). This $\kappa_{org}$ value is lower than that reported by Ruehl et al. (where $\kappa_{org} = 0.5$), although

those measurements were performed on 150 nm particles at $f_{org} = 0.963$. If we apply the compressed film

model (using the parameters established in (Ruehl et al., 2016) for 150 nm particles at $f_{org} = 0.963$) to predict

the critical supersaturation as a function of $f_{org}$, we obtain a dependence that shows a peak $SS_{crit}$ at $f_{org} = 0.4$

(Figure 2), followed by a decrease towards higher $f_{org}$. This shape is consistent with the lower value of $SS_{crit}$

reported by Ruehl et al. at $f_{org} = 0.963$. If we use the ideal molar volume derived $\kappa_{org}$ value of 0.131 and a

constant surface tension of $\sigma = \sigma_w$, the bulk solubility prediction significantly overestimates $SS_{crit}$ for $f_{org} >$

0.5. It must be noted here that without prior knowledge that suberic acid is inherently of very low

hygroscopicity and exhibits low water-solubility, the prediction using $\kappa_{org} = 0.35$ could be mistaken as the

correct answer. However, a $\kappa_{org}$ value of this magnitude is unphysical when considering the original

definition of $\kappa$. One could ignore the physical meaning of $\kappa$ and simply use it as an all-encompassing

parameter to describe activation efficiency. In this case, the generality of the parameter to interpret

observations in different conditions is lost. The compressed film model decouples the water activity

component from the surface tension component, and thus should better represent the physical processes at

work. However, the agreement for all these models breaks down further when looking at different sized

particles. For example, Figure 2B shows the same system of ammonium sulfate and suberic acid, this time

using 40 nm dry diameter particles. In this case, the required supersaturation decreases with increasing

organic fraction, suggesting that suberic acid is more hygroscopic than ammonium sulfate (requiring $\kappa_{org} =$

0.72). For the ultrafine aerosol size reported here, the compressed film model does a worse job at predicting

the behavior, suggesting that it too suffers from a lack of generality in its applicability. What is clear,

however, is that a specific suberic acid hygroscopicity alone cannot explain the observations across a range of particle sizes and compositions.

These observations are obscured when the data is reported with a fixed inorganic seed size with an increasing organic fraction achieved through an increase in the coated diameter. The data as a function of $f_{org}$ with a fixed total diameter was used to plot $SS_{crit}$ as a function of $f_{org}$ with a fixed inorganic seed size,

shown in Figure 2C with an arbitrarily chosen inorganic seed size of 33 nm dry diameter. The value of $SS_{crit}$ for coated diameters 40, 50 and 100 nm was found by linear interpolation of the data in Figures 2A and 2B (and the 50 nm case in Figure 3). The data is compared against a bulk $\kappa$-Köhler prediction, which exhibits a similar trend, although with a slightly smaller slope when $\kappa_{org} = 0.15$. These data are brought to agreement using $\kappa_{org} = 0.4$. However, we have already shown that for the 40 nm case, a value of $\kappa_{org}$ greater than that

of AS is required to explain the data as a function of $f_{org}$ with a fixed dry diameter. Specifically, in Fig. 2C, the organic volume fractions are distinctly different for the three points shown: $f_{org} = 0.40$, 0.70, and 0.96 for the 40, 50, and 100 nm coated diameter cases, respectively. Thus, the choice of experimental procedure or data presentation can impact the interpretation of experimental observations and we caution care when presenting such data. When composition and particle size are coupled in $D_{coated}$, the sensitivity to surface

tension effects is diminished, as smaller particles will contain a lower volume fraction of organic material.

For both sets of measurements, we also applied three model predictions based on the Aerosol Inorganic-Organic Mixtures Functional groups Activity Coefficients (AIOMFAC) (Zuend et al., 2008; 2011) model with LLPS and a phase-specific surface tension mixing rule considered. The full equilibrium calculation, labeled as AIOMFAC-EQUIL in Figure 2, considers the potential existence of a bulk liquid–liquid

equilibrium, resulting in two liquid phases of distinct compositions yet each containing some amounts of all three components. The chemical compositions affect the surface tensions of the individual phases and, using a core-shell morphology assumption and minimum phase (film) thickness, that of the overall droplet. For the calculations performed, suberic acid is assumed to be in a liquid state at high water activity. This model predicts an LLPS for the aqueous suberic acid + ammonium sulfate system, but only up to a certain

water activity level < 0.99, beyond which a single liquid phase is the stable state. The upper limit of LLPS

predicted increases with the fraction of suberic acid in the system: LLPS onset $a_w \approx 0.942$ for $f_{org} = 0.27$ to

$a_w \approx 0.983$ for $f_{org} = 0.88$. This results in the absence of LLPS at supersaturated conditions prior to CCN

activation for both dry particle diameters considered. Therefore, the AIOMFAC-EQUIL prediction does

not lead to a significant surface tension reduction here, which explains why the predicted $SS_{crit}$ in Fig. 2 is

similar to that of an iso-σ κ-Köhler model with $\kappa_{org} \approx 0.13$. The two AIOMFAC-CLLPS variants represent

simplified model calculations in which the assumption is made that dissolved aqueous electrolytes and

organics always reside in separate phases regardless of water content. In the variant labeled AIOMFAC-

CLLPS (w/ org film), all organic material is assumed to reside in a water-free organic shell-phase (an

organic film) at the surface of the aqueous droplet. This assumption leads to a maximum possible surface

tension lowering up to relatively large droplet sizes (for intermediate to high $f_{org}$), yet a reduced solute

effect, especially for high $f_{org}$. The variant labelled AIOMFAC-CLLPS (w/o org film) in Fig. 2 differs by

allowing water to partition to the organic-rich shell phase (in equilibrium with the target water activity),

which may affect the surface tension of that phase. Due to a significant water uptake by suberic acid,

predicted to occur for $a_w > 0.99$, the resulting surface tension prior to and near activation is that of pure

water and the $SS_{crit}$ prediction resembles that of the AIOMFAC-EQUIL case. Physical parameters used in

these simulations are presented in Table 1 and a schematic representation of these cases is shown in Figure

A1. A comparison of predicted Köhler curves from these model variants is shown in Fig. A2 for the case

of $f_{org} = 0.58$. The AIOMFAC-based predictions of critical dry diameters and $SS_{crit}$ are listed in Table 2 for

a range of dry diameters. A comparison of the different models with experimental data in Fig. 2 indicate

that for $f_{org} > 0.3$, the simplified organic film model variant AIOMFAC-CLLPS (w/ org film) offers the best

agreement with the measurements. This observation is consistent with the results of (Prisle et al., 2011)

who applied a similar simple model to droplets containing ionic surfactants and hints at a significant

suppression of surface tension by suberic acid, which is likely highly enriched at the droplet surface.

Unfortunately, neither of the models fully captures the observed behavior at all $f_{org}$ and size regimes, and is suggestive that additional factors that have yet to be fully identified may influence the activation process. As discussed by Ovadnevaite and coworkers (Ovadnevaite et al., 2017), the AIOMFAC-EQUIL and AIOMFAC-CLLPS (w/ org film) calculations may provide upper and lower bounds on the prediction of $SS_{crit}$ for a given system, which is roughly in agreement with the data in Fig. 2. In reality, it is likely that some portion of suberic acid dissolves into the aqueous droplet bulk at high relative humidity, itself contributing to the water uptake of the droplet, as predicted by AIOMFAC-EQUIL, while a significant organic enrichment prevails at the surface, lowering the surface tension and consequently $SS_{crit}$. Such behavior could explain the data and the increasing model-measurement deviations towards higher $f_{org}$. Improvements of the AIOMFAC-based models with more sophisticated bulk–surface partitioning treatments in individual liquid phases seem to offer a way forward to address some of the observed shortcomings in future work. At this point, it remains intriguing that the simplified organic film model provides the best description of these experimental data, even though its restrictive assumptions about phase separation and organic water content seem to make it a less physically realistic model variant.

These types of experiment also expose further factors that influence CCN activation, possibly through modification to surface partitioning, such as the role of inorganic ions. We performed additional measurements using different inorganic seed particles coated with suberic acid and observed vastly different behavior across three different salts (ammonium sulfate, sodium iodide and sodium carbonate), shown in Figure 3 for 50 nm dry diameter particles. We see for ammonium sulfate the same qualitative behavior as for the other two sizes already discussed; in contrast, the responses of the systems containing the other salts (all with suberic acid as the organic component) are very different. Sodium carbonate exhibits an increase in the required critical supersaturation across the range of compositions; a trend that could reasonably be predicted without invoking surface tension effects. The trend with sodium iodide is more complex and appears to show a sharp discontinuity near $f_{org} = 0.5$, which was highly reproducible across multiple repeat experiments over multiple days. These salts were chosen to span the range of the Hofmeister series, which

describes the propensity of inorganic ions to salt in or salt out organic molecules (proteins in particular).

Sulfate and carbonate are the best salting-out ions, while iodide has a relatively weak salting-out effect due

to its own surface propensity (Santos et al., 2010). It is interesting to note the differences between carbonate

and sulfate, despite their similar position on the Hofmeister scale. The role of the cation is generally

considered to be much smaller than that of the anion, thus these differences are non-trivial. These results

serve to further highlight a key conclusion of this work – that we currently lack a robust molecular model

that is capable of describing and therefore accurately predicting CCN hygroscopicity and activation even

in a relatively simple model system. We hope to prompt further discussions and experimental studies to

explore these observations and bulk/surface composition effects on surface tension and CCN activation in

more detail.

**5 Summary and Conclusions**

Surface tension effects can lead to significant differences from classic, hygroscopicity mixing rule

mechanisms for CCN activation (Hansen et al., 2015; Kristensen et al., 2014). While it has already been

made clear that the activation diameter can be significantly different from that determined by an *iso-σ*

Köhler curve, in this work we reveal the potential for more subtle changes in CCN activity (both increases

and decreases relative to pure ammonium sulfate particles) as a result of the organics-influenced surface

tension evolution during droplet growth. These changes were captured by measuring particles at a fixed

diameter with a range of organic volume fractions. Ultimately, to derive an accurate picture of CCN activity

across the relevant ranges of chemical compositions and size distributions, the effects of surface tension

variability must be taken into account. It should be noted, however, that there are many situations where

using simple mixing rules with inferred values for $\kappa_{org}$ can lead to sufficiently accurate predictions without

the need for more complex analyses or simulations. It is therefore of key importance to constrain the

conditions under which simple approaches are justified – and to know when they may be inappropriate.

Taking the activation model based on Köhler theory forward will require a more rigorous interrogation of

the role of co-solutes in partitioning, and ultimately an assessment of its effect in real-world simulations of cloud formation.

In the meantime, it is important for environmental scientists to recognize the conditions in which surface effects may be influencing cloud droplet formation, e.g., low solubility or insoluble organics mixed with inorganic salts, high-RH phase separation, small particle sizes with critical supersaturations close to the peak supersaturations experienced in clouds, etc. We suggest, if possible, that experimental data be explored as a function of organic volume fraction while keeping the overall dry particle size the same, as from our

laboratory experiments this dependence shows the most clear indicator of an unexplained size effect that may be attributed to bulk–surface partitioning. In experiments where both size and composition vary, the contribution from each is less clear and the effect of the organic component due to bulk–surface partitioning could be hidden. Further fundamental laboratory and modeling studies being performed will allow for in-depth testing and refinements of the proposed models and mechanisms that describe bulk–surface

partitioning and surface tension, ultimately reaching towards a robust and universal mechanism that allows both hygroscopicity and surface tension effects to be coupled into a practical framework.

**Acknowledgements**

AZ acknowledges the support of Natural Sciences and Engineering Research Council of Canada (NSERC), through grant RGPIN/04315-2014. Work on this topic by KRW is supported by the Condensed Phase and

Interfacial Molecular Science Program, in the Chemical Sciences Geosciences and Biosciences Division of the Office of Basic Energy Sciences of the U.S. Department of Energy under Contract No. DE-AC02-05CH11231.

**Table 1: Value of physical parameters used in AIOMFAC calculations. Density of mixtures were calculated as a linear (additive) combination of the apparent molar volumes of the contributions of water, ammonium sulfate (AS), and suberic acid.**

| Calculation parameter | Value | Unit |
|---|---|---|
| Temperature, T | 293.15 | K |
| Pure comp. surface tension water (at T) (Vargaftik et al., 1983) | 72.75 | mJ m$^{-2}$ |
| Pure comp. surface tension Suberic Acid (at T) * | 35.00 | mJ m$^{-2}$ |
| Surface tension of aqueous AS (at T) ** | 72.75 | mJ m$^{-2}$ |
| Density of pure water (liq.) at T | 997 | kg m$^{-3}$ |
| Density of pure suberic acid (liq.) at T | 1220 | kg m$^{-3}$ |
| Density of pure AS (liq.) at T (Clegg and Wexler, 2011) | 1550 | kg m$^{-3}$ |
| Density of pure AS (solid) at T (Clegg and Wexler, 2011) | 1770 | kg m$^{-3}$ |
| Minimum shell phase thickness, $\delta_{\beta,min}$ | 0.3 | nm |

*Value taken from measurements for of adipic acid (Riipinen et al., 2007) on the basis of structural similarity to suberic acid. **Assumption of no influence on droplet surface tension compared to water here (since highly dilute).

**Table 2: Critical supersaturation $SS_{crit}$ (%) for given dry diameters ($D_{dry,crit}$) and organic volume fractions ($f_{org}$) in dry particle at $T$ = 293.15 K; predicted by the different AIOMFAC-based models.**

AIOMFAC-EQUIL; with liquid–liquid phase separation considered when predicted

| Mixture | | Dry diameter [nm] (of overall particle) | | | | | | | | | | | |
|---|---|---|---|---|---|---|---|---|---|---|---|---|---|
| Solutes | $f_{org}$ | 30 | 35 | 40 | 45 | 50 | 60 | 80 | 100 | 120 | 140 | 160 | 200 |
| AS | 0.00 | 1.005 | 0.788 | 0.639 | 0.532 | 0.451 | 0.339 | 0.217 | 0.154 | 0.116 | 0.091 | 0.074 | 0.053 |
| Suberic + AS | 0.27 | 1.135 | 0.893 | 0.726 | 0.605 | 0.513 | 0.387 | 0.248 | 0.176 | 0.133 | 0.105 | 0.085 | 0.061 |
| Suberic + AS | 0.40 | 1.214 | 0.957 | 0.779 | 0.649 | 0.552 | 0.416 | 0.267 | 0.190 | 0.143 | 0.113 | 0.092 | 0.066 |
| Suberic + AS | 0.58 | 1.350 | 1.068 | 0.871 | 0.727 | 0.619 | 0.468 | 0.301 | 0.214 | 0.162 | 0.128 | 0.105 | 0.074 |
| Suberic + AS | 0.75 | 1.532 | 1.214 | 0.993 | 0.831 | 0.708 | 0.537 | 0.347 | 0.247 | 0.187 | 0.148 | 0.121 | 0.086 |
| Suberic + AS | 0.88 | 1.751 | 1.389 | 1.136 | 0.951 | 0.811 | 0.616 | 0.399 | 0.285 | 0.216 | 0.172 | 0.140 | 0.100 |

AIOMFAC-CLLPS (w/ org film); organic phase assumed water-free

| Solutes | $f_{org}$ | 30 | 35 | 40 | 45 | 50 | 60 | 80 | 100 | 120 | 140 | 160 | 200 |
|---|---|---|---|---|---|---|---|---|---|---|---|---|---|
| Suberic + AS | 0.27 | 0.972 | 0.766 | 0.624 | 0.520 | 0.442 | 0.334 | 0.215 | 0.153 | 0.116 | 0.091 | 0.075 | 0.053 |
| Suberic + AS | 0.40 | 0.961 | 0.759 | 0.618 | 0.516 | 0.439 | 0.333 | 0.214 | 0.153 | 0.116 | 0.092 | 0.075 | 0.053 |
| Suberic + AS | 0.58 | 0.950 | 0.751 | 0.613 | 0.513 | 0.437 | 0.331 | 0.214 | 0.153 | 0.116 | 0.092 | 0.075 | 0.054 |
| Suberic + AS | 0.75 | 0.944 | 0.747 | 0.611 | 0.511 | 0.436 | 0.331 | 0.214 | 0.153 | 0.116 | 0.092 | 0.076 | 0.054 |
| Suberic + AS | 0.88 | 0.987 | 0.781 | 0.637 | 0.532 | 0.453 | 0.342 | 0.220 | 0.156 | 0.118 | 0.093 | 0.076 | 0.054 |

AIOMFAC-CLLPS (w/o org film); water uptake by organic-rich phase considered

| Solutes | $f_{org}$ | 30 | 35 | 40 | 45 | 50 | 60 | 80 | 100 | 120 | 140 | 160 | 200 |
|---|---|---|---|---|---|---|---|---|---|---|---|---|---|
| Suberic + AS | 0.27 | 1.108 | 0.878 | 0.716 | 0.598 | 0.509 | 0.385 | 0.247 | 0.176 | 0.133 | 0.105 | 0.085 | 0.061 |
| Suberic + AS | 0.40 | 1.197 | 0.948 | 0.774 | 0.646 | 0.550 | 0.416 | 0.267 | 0.190 | 0.144 | 0.113 | 0.092 | 0.066 |
| Suberic + AS | 0.58 | 1.357 | 1.075 | 0.877 | 0.733 | 0.624 | 0.472 | 0.303 | 0.216 | 0.163 | 0.129 | 0.105 | 0.075 |
| Suberic + AS | 0.75 | 1.579 | 1.246 | 1.017 | 0.849 | 0.723 | 0.547 | 0.352 | 0.250 | 0.189 | 0.150 | 0.122 | 0.087 |
| Suberic + AS | 0.88 | 1.999 | 1.592 | 1.303 | 1.089 | 0.926 | 0.697 | 0.443 | 0.301 | 0.220 | 0.174 | 0.142 | 0.101 |

**Table 3: Experimentally measured critical supersaturation for given dry diameter, inorganic particle core, and organic volume fraction.**

| $D_{dry}$ / nm | Inorganic | $f_{org}$ | $SS_{crit}$ (%) |
|---|---|---|---|
| 100 | $(NH_4)_2SO_4$ | 0.00 | 0.16 |
| | | 0.27 | 0.17 |
| | | 0.58 | 0.17 |
| | | 0.88 | 0.19 |
| 40 | $(NH_4)_2SO_4$ | 0.00 | 0.66 |
| | | 0.33 | 0.62 |
| | | 0.58 | 0.57 |
| | | 0.88 | 0.60 |
| 50 | $(NH_4)_2SO_4$ | 0.00 | 0.45 |
| | | 0.27 | 0.45 |
| | | 0.41 | 0.42 |
| | | 0.66 | 0.41 |
| | | 0.88 | 0.39 |
| 50 | NaI | 0.00 | 0.38 |
| | | 0.27 | 0.41 |
| | | 0.41 | 0.43 |
| | | 0.49 | 0.45 |
| | | 0.53 | 0.41 |
| | | 0.56 | 0.41 |
| | | 0.66 | 0.42 |
| | | 0.78 | 0.45 |
| | | 0.88 | 0.48 |
| 50 | $Na_2CO_3$ | 0.00 | 0.33 |
| | | 0.27 | 0.34 |
| | | 0.49 | 0.35 |
| | | 0.66 | 0.41 |
| | | 0.78 | 0.42 |
| | | 0.88 | 0.44 |

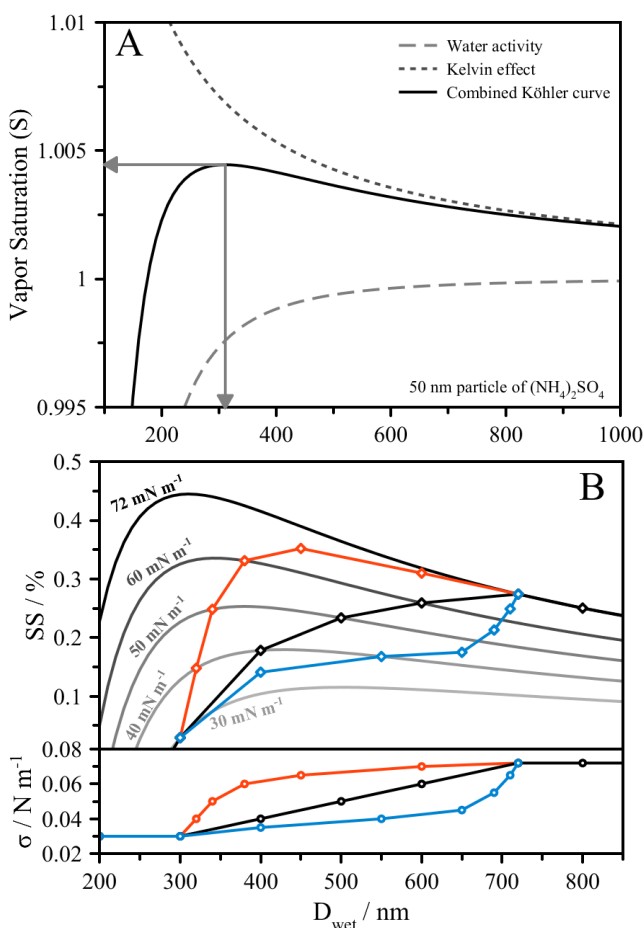

**Figure 1: (A)** Köhler curve construction from the combination of the water activity term and the Kelvin effect, shown here for 50 nm particles of ammonium sulfate. The arrows indicate the critical supersaturation ($SS_{crit}$) and the critical wet activation diameter ($D_{crit}$). **(B)** Köhler curves (NB. $SS = (S\text{-}1) \times 100\,\%$) of varying fixed surface tension values for 50 nm (dry diameter) particles with water activity treated as an ammonium sulfate solution. A schematic linear dependence of surface tension on droplet diameter is shown in black, and the Köhler curve construction that takes into account the change in surface tension is shown in bold and with diamond symbols. Additional surface tension dependencies are shown in red, which exhibits activation at $\sigma < \sigma_w$, and in blue, which shows a dramatic increase in the critical wet diameter.



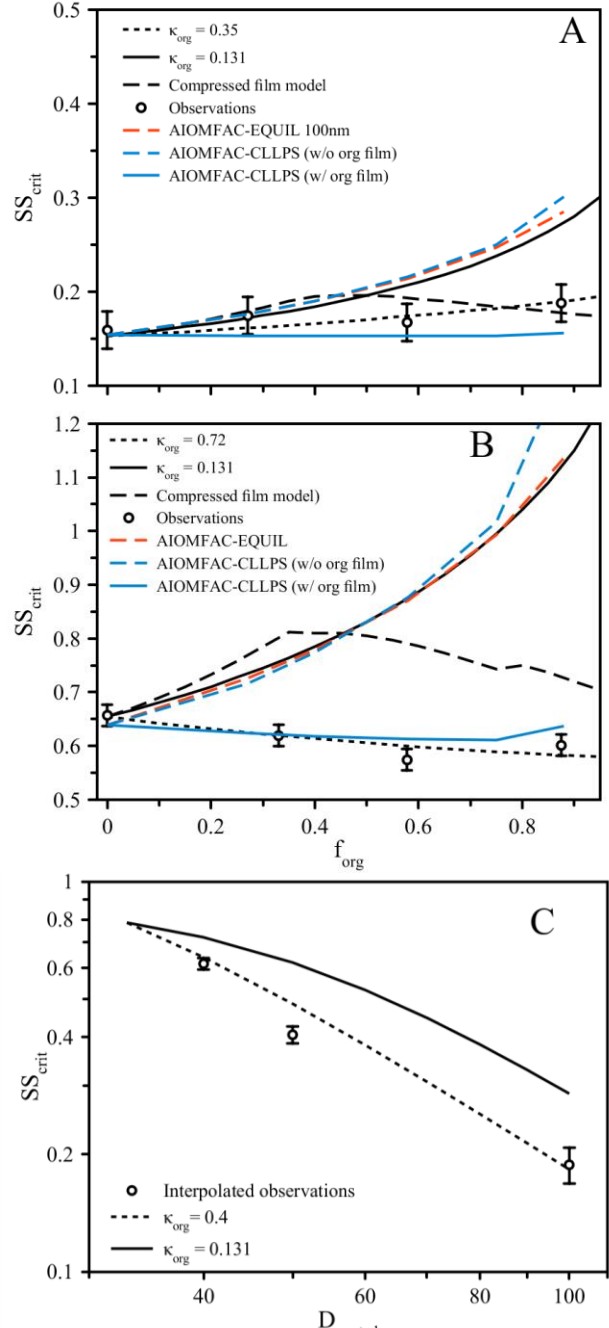

**Figure 2: (A) Measured critical supersaturation ($SS_{crit}$) of size-selected ammonium sulfate particles coated with suberic acid (points) to a set dry diameter of 100 nm at $T \approx 293$ K. The curves show predictions from the compressed film model of Ruehl et al., a simple $\kappa$-Köhler model with constant surface tension, and model variants from the AIOMFAC-based framework (see text). (B) Analogous to (A) but for particles of ammonium sulfate coated to 40 nm dry diameter by suberic acid. (C) Using the data from panels A and B, $SS_{crit}$ is shown as a function of coated diameter with a fixed inorganic seed of 33 nm (points). The lines indicate the $\kappa$-Köhler model with constant surface tension using different but constant values of $\kappa_{org}$.**

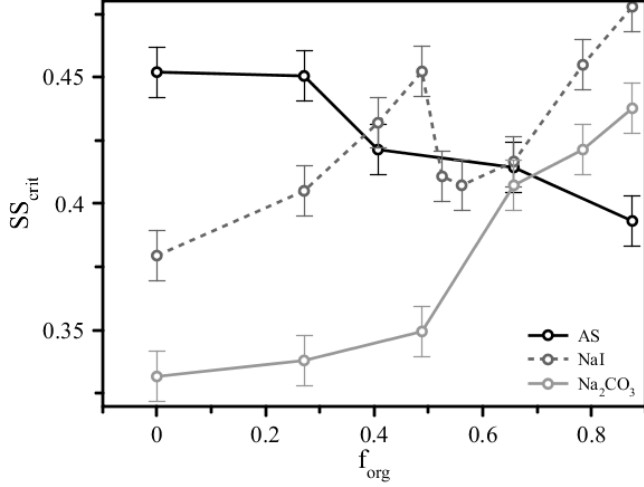

**Figure 3: The measured critical supersaturation for 50 nm dry diameter particles comprised of different salts (ammonium sulfate, sodium iodide and sodium carbonate) and variable volume fractions of suberic acid ($f_{org}$) at $T \approx 293$ K.**


**Appendix**

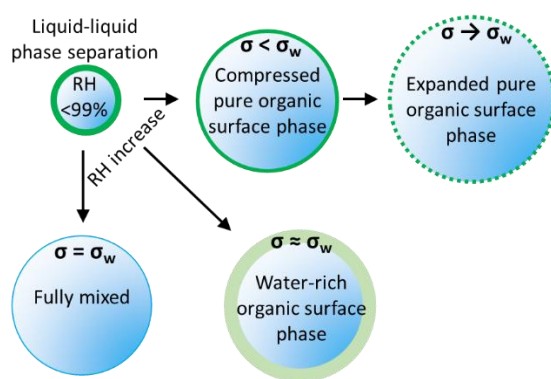


**Figure A1: Schematic representation of the three model variants. AIOMFAC-EQUIL treats the droplets as a fully mixed single phase, with the surface tension of the wet droplet sharply approaching $\sigma_w$. The phase separated variants considered a surface phase that is water-rich, with surface tension close to $\sigma_w$, and a surface phase that excludes water, behaving as an organic surface film, with $\sigma < \sigma_w$.**

**The output from these scenarios are shown in Figure A2.**

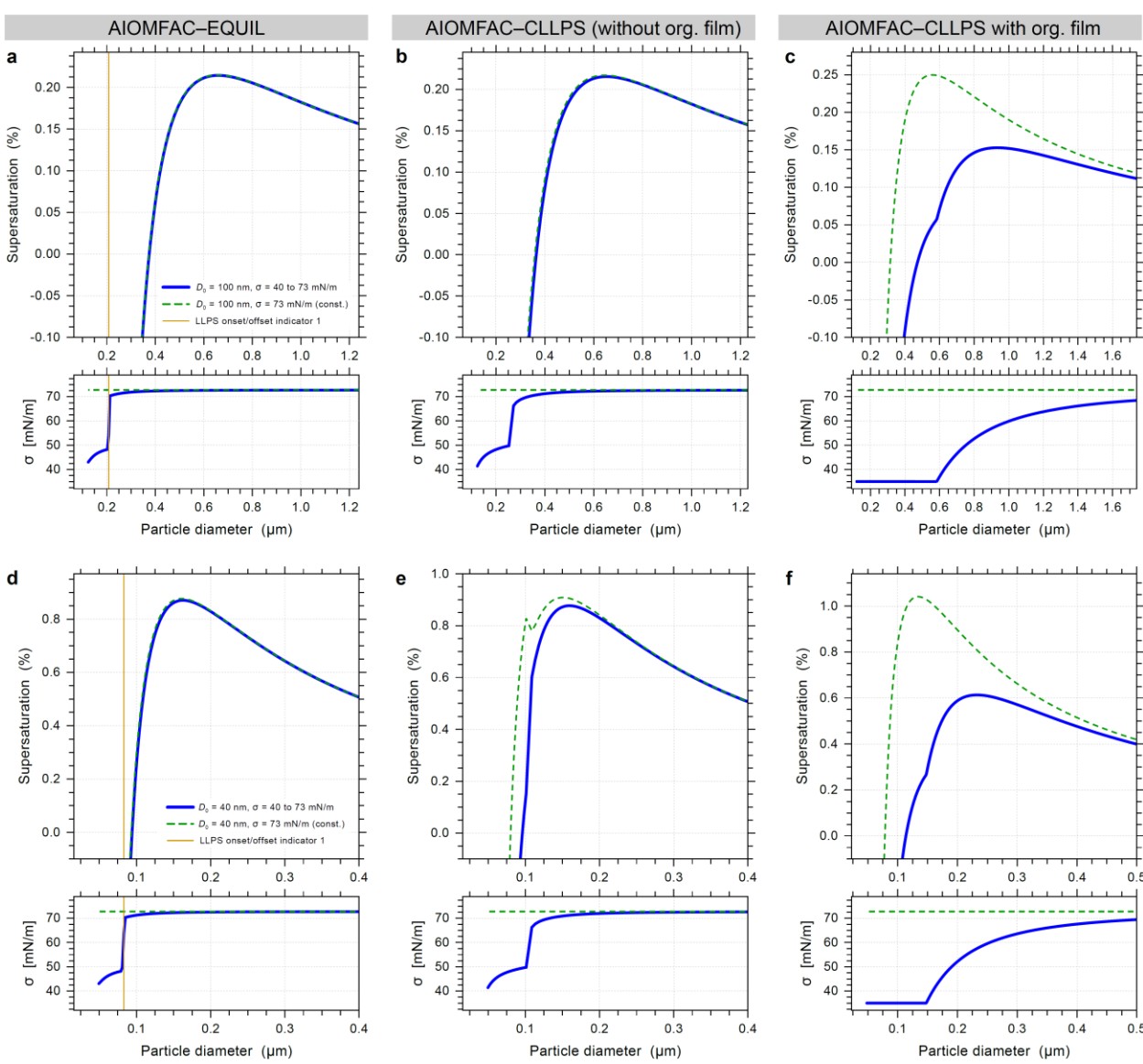

**Figure A2: Predicted Köhler curves and associated droplet surface tensions for 100 nm (a, b, c) and 40 nm (d, e, f) dry diameter particles of ammonium sulfate with suberic acid at $f_{org} = 0.58$ using the AIOMFAC models described in the main text. The AIOMFAC-CLLPS with org. film gives the closest predictions to the experimental observations and predicts similar shaped curves to those measured experimentally by (Ruehl et al., 2016).**

Abdul-Razzak, H. and Ghan, S. J.: A parameterization of aerosol activation: 2. Multiple aerosol types, J Geophys Res

Atmospheres, 105(D5), 6837–6844, doi:10.1029/1999jd901161 , 2000.

Asa-Awuku, A. and Nenes, A.: Effect of solute dissolution kinetics on cloud droplet formation: Extended Köhler theory, J

Geophys Res Atmospheres 1984 2012, 112(D22201), D22201, doi:10.1029/2005jd006934 , 2007.

Asa-Awuku, A., Sullivan, A., Hennigan, C., Weber, R. and Nenes, A.: Investigation of molar volume and surfactant

characteristics of water-soluble organic compounds in biomass burning aerosol, Atmos Chem Phys, 8(4), 799--812,

doi:10.5194/acp-8-799-2008 , 2008.

Asa-Awuku, A., Engelhart, G., Lee, B., P, is, S. and Nenes, A.: Relating CCN activity, volatility, and droplet growth kinetics of

$\beta$-caryophyllene secondary organic aerosol, Atmos Chem Phys, 9(3), 795--812, doi:10.5194/acp-9-795-2009 , 2009.

Bilde, M. and Svenningsson, B.: CCN activation of slightly soluble organics: the importance of small amounts of inorganic salt

and particle phase, Tellus B, 56(2), doi:10.3402/tellusb.v56i2.16406 , 2017.

Boyer, H. C. and Dutcher, C. S.: Atmospheric Aqueous Aerosol Surface Tensions: Isotherm-based Modeling and Biphasic

Microfluidic Measurements, J Phys Chem, doi:10.1021/acs.jpca.7b03189 , 2017.

Boyer, H. C., Bzdek, B., Reid, J. and Dutcher, C. S.: A Statistical Thermodynamic Model for Surface Tension of Organic and

Inorganic Aqueous Mixtures, J Phys Chem, 121(1), 198–205, doi:10.1021/acs.jpca.6b10057 , 2016.


Collins, D. B., Bertram, T. H., Sultana, C. M., Lee, C., Axson, J. L. and Prather, K. A.: Phytoplankton blooms weakly influence

the cloud forming ability of sea spray aerosol, Geophys Res Lett, 43(18), 9975--9983, doi:10.1002/2016gl069922 , 2016.

Facchini, M., Decesari, S., Mircea, M., Fuzzi, S., ro and Loglio, G.: Surface tension of atmospheric wet aerosol and cloud/fog

droplets in relation to their organic carbon content and chemical composition, Atmos Environ, 34(28), 4853--4857,

doi:10.1016/s1352-2310(00)00237-5 , 2000.

Forestieri, S. D., Staudt, S. M., Kuborn, T. M., Faber, K., Ruehl, C. R., Bertram, T. H. and Cappa, C. D.: Establishing the impact

of model surfactants on cloud condensation nuclei activity of sea spray aerosol mimics, Atmos Chem Phys, 18(15), 10985–
11005, doi:10.5194/acp-18-10985-2018 , 2018.

Frosch, M., Prisle, N., Bilde, M., Varga, Z. and Kiss, G.: Joint effect of organic acids and inorganic salts on cloud droplet

activation, Atmos Chem Phys, 11(8), 3895--3911, doi:10.5194/acp-11-3895-2011 , 2011.

Fuentes, E., Coe, H., Green, D. and McFiggans, G.: On the impacts of phytoplankton-derived organic matter on the properties of

the primary marine aerosol - Part 2: Composition, hygroscopicity and cloud condensation activity, Atmos Chem Phys, 11(6),

2585--2602, doi:10.5194/acp-11-2585-2011 , 2011.

Good, N., Topping, D., Allan, J., Flynn, M., Fuentes, E., Irwin, M., Williams, P., Coe, H. and McFiggans, G.: Consistency

between parameterisations of aerosol hygroscopicity and CCN activity during the RHaMBLe discovery cruise, Atmos Chem

Phys, 10(7), 3189--3203, doi:10.5194/acp-10-3189-2010 , 2010.

Hings, S., Wrobel, W., Cross, E., Worsnop, D., Davidovits, P. and Onasch, T.: CCN activation experiments with adipic acid:

Effect of particle phase and adipic acid coatings on soluble and insoluble particles, Atmos Chem Phys, 8(14), 3735--3748,

doi:10.5194/acp-8-3735-2008 , 2008.

Jura, G. and Harkins, W. D.: Surfaces of Solids. XIV. A Unitary Thermodynamic Theory of the Adsorption of Vapors on Solids

and of Insoluble Films on Liquid Subphases, J Am Chem Soc, 68(10), 1941–1952, doi:10.1021/ja01214a022 , 1946.

Köhler, H.: The nucleus in and the growth of hygroscopic droplets, T Faraday Soc, 32(0), 1152–1161, doi:10.1039/tf9363201152

, 1936.

Kuwata, M., Shao, W., Lebouteiller, R. and Martin, S.: Classifying organic materials by oxygen-to-carbon elemental ratio to

predict the activation regime of Cloud Condensation Nuclei (CCN), Atmos Chem Phys, 13(10), 5309--5324, doi:10.5194/acp-13-

580    5309-2013 , 2013.

Li, Z., Williams, A. L. and Rood, M. J.: Influence of Soluble Surfactant Properties on the Activation of Aerosol Particles

Containing Inorganic Solute, J Atmos Sci, 55(10), 1859–1866, doi:10.1175/1520-0469(1998)055<1859:iosspo>2.0.co;2 , 1998.

McFiggans, G., Artaxo, P., Baltensperger, U., Coe, H., Facchini, M., Feingold, G., Fuzzi, S., Gysel, M., Laaksonen, A., Lohmann, U., Mentel, T., Murphy, D., O'Dowd, C., Snider, J. and Weingartner, E.: The effect of physical and chemical aerosol properties on warm cloud droplet activation, Atmos Chem Phys, 6(9), 2593–2649, doi:10.5194/acp-6-2593-2006 , 2006.

Nguyen, Q. T., Kjær, K. H., Kling, K. I., Boesen, T. and Bilde, M.: Impact of fatty acid coating on the CCN activity of sea salt
particles, Tellus B Chem Phys Meteorology, 69(1), 1304064, doi:10.1080/16000889.2017.1304064 , 2017.

Nozière, B., Baduel, C. and Jaffrezo, J.-L.: The dynamic surface tension of atmospheric aerosol surfactants reveals new aspects of cloud activation, Nat Commun, 5(1), 3335, doi:10.1038/ncomms4335 , 2014.

Ovadnevaite, J., Ceburnis, D., Martucci, G., Bialek, J., Monahan, C., Rinaldi, M., Facchini, M., Berresheim, H., Worsnop, D. R. and O'Dowd, C.: Primary marine organic aerosol: A dichotomy of low hygroscopicity and high CCN activity, Geophys Res Lett, 38(21), n/a-n/a, doi:10.1029/2011gl048869 , 2011.

Ovadnevaite, J., Zuend, A., Laaksonen, A., Sanchez, K. J., Roberts, G., Ceburnis, D., Decesari, S., Rinaldi, M., Hodas, N.,
Facchini, M., Seinfeld, J. H. and Dowd, C.: Surface tension prevails over solute effect in organic-influenced cloud droplet activation, Nature, 546(7660), 637, doi:10.1038/nature22806 , 2017.

Petters, M. and Kreidenweis, S.: A single parameter representation of hygroscopic growth and cloud condensation nucleus activity, Atmos Chem Phys, 7(8), 1961–1971, doi:10.5194/acp-7-1961-2007 , 2007.
Petters, M. and Kreidenweis, S.: A single parameter representation of hygroscopic growth and cloud condensation nucleus activity – Part 2: Including solubility, Atmos Chem Phys, 8(20), 6273–6279, doi:10.5194/acp-8-6273-2008 , 2008.

Petters, M. and Kreidenweis, S.: A single parameter representation of hygroscopic growth and cloud condensation nucleus
activity – Part 3: Including surfactant partitioning, Atmos Chem Phys, 13(2), 1081--1091, doi:10.5194/acp-13-1081-2013 , 2013.

Petters, M., Kreidenweis, S. and Ziemann, P.: Prediction of cloud condensation nuclei activity for organic compounds using functional group contribution methods, Geosci Model Dev, 9(1), 111–124, doi:10.5194/gmd-9-111-2016 , 2016.

Petters, S. and Petters, M.: Surfactant effect on cloud condensation nuclei for two-component internally mixed aerosols, J

Geophys Res Atmosferes, 121(4), 1878–1895, doi:10.1002/2015jd024090 , 2016.

Prisle, N., Raatikainen, T., Laaksonen, A. and Bilde, M.: Surfactants in cloud droplet activation: mixed organic-inorganic particles, Atmos Chem Phys, 10(12), 5663–5683, doi:10.5194/acp-10-5663-2010 , 2010.


Prisle, N., Maso, D. M. and Kokkola, H.: A simple representation of surface active organic aerosol in cloud droplet formation, Atmos Chem Phys, 11(9), 4073–4083, doi:10.5194/acp-11-4073-2011 , 2011.

Prisle, N. L., Raatikainen, T., Sorjamaa, Svenningsson, B., Laaksonen, A. and Bilde, M.: Surfactant partitioning in cloud droplet

activation: a study of C8, C10, C12 and C14 normal fatty acid sodium salts, Tellus B, 60(3), 416–431, doi:10.1111/j.1600-0889.2008.00352.x , 2008.

Rastak, N., Pajunoja, A., Navarro, A. J., Ma, J., Song, M., Partridge, D., Kirkevåg, A., Leong, Y., Hu, W., Taylor, N., Lambe, A., Cerully, K., Bougiatioti, A., Liu, P., Krejci, R., Petäjä, T., Percival, C., Davidovits, P., Worsnop, D., Ekman, A., Nenes, A.,

Martin, S., Jimenez, J., Collins, D., Topping, D. O., Bertram, A., Zuend, A., Virtanen, A. and Riipinen, I.: Microphysical explanation of the RH-dependent water affinity of biogenic organic aerosol and its importance for climate, Geophys Res Lett, 44(10), 5167–5177, doi:10.1002/2017gl073056 , 2017.

Renbaum-Wolff, L., Song, M., Marcolli, C., Zhang, Y., Liu, P. F., Grayson, J. W., Geiger, F. M., Martin, S. T. and Bertram, A.

K.: Observations and implications of liquid–liquid phase separation at high relative humidities in secondary organic material produced by α-pinene ozonolysis without inorganic salts, Atmos Chem Phys, 16(12), 7969–7979, doi:10.5194/acp-16-7969-2016 , 2016.

Riipinen, I., Koponen, I. K., Frank, G. P., Hyvärinen, A.-P., Vanhanen, J., Lihavainen, H., Lehtinen, K. E., Bilde, M. and

Kulmala, M.: Adipic and Malonic Acid Aqueous Solutions:  Surface Tensions and Saturation Vapor Pressures, J Phys Chem, 111(50), 12995–13002, doi:10.1021/jp073731v , 2007.

Roberts, G. and Nenes, A.: A Continuous-Flow Streamwise Thermal-Gradient CCN Chamber for Atmospheric Measurements, Aerosol Sci Tech, 39(3), 206–221, doi:10.1080/027868290913988 , 2010.


Ruehl, C. R. and Wilson, K. R.: Surface Organic Monolayers Control the Hygroscopic Growth of Submicrometer Particles at

High Relative Humidity, J Phys Chem, 118(22), 3952–3966, doi:10.1021/jp502844g , 2014.

Ruehl, C. R., Davies, J. F. and Wilson, K. R.: An interfacial mechanism for cloud droplet formation on organic aerosols, Science,

351(6280), 1447–1450, doi:10.1126/science.aad4889 , 2016.

Santos, A., re , P., Diehl, A., re and Levin, Y.: Surface tensions, surface potentials, and the hofmeister series of electrolyte

solutions, Langmuir, 26(13), 10778--10783, doi:10.1021/la100604k , 2010.

Shulman, M. L., Jacobson, M. C., Carlson, R. J., Synovec, R. E. and Young, T. E.: Dissolution behavior and surface tension

effects of organic compounds in nucleating cloud droplets, Geophys Res Lett, 23(3), 277–280, doi:10.1029/95gl03810 , 1996.

Sorjamaa, R., Svenningsson, B., Raatikainen, T., Henning, S., Bilde, M. and Laaksonen, A.: The role of surfactants in Köhler

theory reconsidered, Atmos Chem Phys, 4(8), 2107–2117, doi:10.5194/acp-4-2107-2004 , 2004.


Svenningsson, B., Rissler, J., Swietlicki, E., Mircea, M., Bilde, M., Facchini, M., Decesari, S., Fuzzi, S., Zhou, J., Mønster, J. and

Rosenørn, T.: Hygroscopic growth and critical supersaturations for mixed aerosol particles of inorganic and organic compounds

of atmospheric relevance, Atmos Chem Phys, 6(7), 1937–1952, doi:10.5194/acp-6-1937-2006 , 2006.

Topping, D. and McFiggans, G.: Tight coupling of particle size, number and composition in atmospheric cloud droplet activation,

Atmos Chem Phys, 12(7), 3253–3260, doi:10.5194/acp-12-3253-2012 , 2012.

Topping, D., McFiggans, G., Kiss, G., Varga, Z., Facchini, M., Decesari, S. and Mircea, M.: Surface tensions of multi-

component mixed inorganic/organic aqueous systems of atmospheric significance: measurements, model predictions and

importance for cloud activation predictions, Atmos Chem Phys, 7(9), 2371–2398, doi:10.5194/acp-7-2371-2007 , 2007.

Topping, D., Connolly, P. and McFiggans, G.: Cloud droplet number enhanced by co-condensation of organic vapours, Nat

Geosci, 6(6), 443, doi:10.1038/ngeo1809 , 2013.

Wang, C., Lei, Y., Endo, S. and Wania, F.: Measuring and Modeling the Salting-out Effect in Ammonium Sulfate Solutions,

Environ Sci Technol, 48(22), 13238–13245, doi:10.1021/es5035602 , 2014.

Yakobi-Hancock, J., Ladino, L., Bertram, A., Huffman, J., Jones, K., Leaitch, W., Mason, R., Schiller, C., Toom-Sauntry, D., Wong, J. and Abbatt, J.: CCN activity of size-selected aerosol at a Pacific coastal location, Atmos Chem Phys, 14(22), 12307–

12317, doi:10.5194/acp-14-12307-2014 , 2014.

Zuend, A., Marcolli, C., Luo, B. and Peter, T.: A thermodynamic model of mixed organic-inorganic aerosols to predict activity coefficients, Atmos Chem Phys, 8(16), 4559–4593, doi:10.5194/acp-8-4559-2008 , 2008.

Zuend, A., Marcolli, C., Booth, A., Lienhard, D., Soonsin, V., Krieger, U., Topping, D., McFiggans, G., Peter, T. and Seinfeld, J.: New and extended parameterization of the thermodynamic model AIOMFAC: calculation of activity coefficients for organic-inorganic mixtures containing carboxyl, hydroxyl, carbonyl, ether, ester, alkenyl, alkyl, and aromatic functional groups, Atmos Chem Phys, 11(17), 9155--9206, doi:10.5194/acp-11-9155-2011 , 2011.