# Peer review of "Technical Note: The Role of Evolving Surface Tension in the Formation of Cloud Droplets"

_Atmospheric Chemistry and Physics, 2018_

## Referee Comment (RC1) · Anonymous Referee #1 · 26 Dec 2018

The technical note summarizes a set of interesting points to be clarified regarding the role of surface active organic aerosol in cloud activation. In light of several recent works, including those of the authors, this topic is very timely and their call for a discussion on the topic is commendable. While it is useful on occasion to summarize the knowledge and open questions in the field, my main concern is that the note brings very little new to the table. Most or all of the points have been made previously, but little references or perspective is given to the substantial body of work already existing on the subject. Some examples are:

p.2 l. 36-38: "In this note, we focus on the role of surface tension, and discuss the limitations of current approximations in light of recently published works that reveal how it is primarily the evolution of surface tension that impacts the activation process." Only

two papers are discussed in detail, the works of Ruehl et al. (2016) and Ovadnevaite et al. (2017). The discussion is not sufficiently substantiated independently of these works.

p.3 l.72-75: Effects of inorganic co-solutes is discussed by e.g. Svenningsson et al. (2006), Prisle et al. (2010,2011), Frosch et al. (2011), Kristensen et al. (2014), and Hansen et al. (2015). Effects of dissolution is discussed by e.g. Shulman et al. (1996) and Bilde and Svenningsson (2004).

p.4. l.81: "...there are many observations that cannot be explained in such simple terms". Please be specific, which observations cannot be explained and what is the argument that they are unexplained.

p. 5-6 and Fig. 1(A): A substantial part of the manuscript is devoted to explaining basic Köhler theory, as done by many authors previously, including basic textbook material. The presentation could benefit from a similarly detailed presentation of the surface tension and partitioning models used.

p.7 l. 171-72: "This also seemingly supports previous assertions that surface tension does not impact activation, since it is generally argued that at the point of activation the droplet is sufficiently dilute and essentially exhibits a surface tension like pure water." Please reference a sufficient amount of such previous assertions to support the claim of a general argument.

p.8 l.180: "... knowing the surface tension only at the point of activation is in sufficient to determine the critical properties at activation..." Please provide references.

p.11 l.258-64: "This predicted size effect indicates that it is not generally valid to assume that all activating CCN will have a surface tension equivalent to or close to that of pure water –nor is it appropriate to use a single measurement of the surface tension of a multicomponent CCN of known dry composition at its activation size (only) to determine its Köhler curve. Furthermore, these model predictions also suggest that

measurements of the surface tension of larger CCN particles (e.g. > 150 nm dry diameter) may not allow for conclusions about the surface tension of much smaller CCN, e.g. of 50 nm dry diameter." These points have certainly been made by other studies than the one specifically described, including those mentioned above, and Wittbom et al. (2018). A broader perspective should be provided.

p.12 l. 292-92: "...in contrast to other studies that allow both coated particle size and organic volume fraction to vary simultaneously." Although in many studies particles are produced by other methods than coating (e.g. nebulization), again many other works have indeed recognized the importance of independently varying the particle size and organic fraction, for studying the effects of organic surface activity and mixing effects, including those listed above.

p.13 l.310-12: "One could ignore the physical meaning of K and simply use it as an all-encompassing parameter to describe activation efficiency." This was already done by several previous studies, e.g. Kristensen et al. (2014) and Hansen et al. (2015). I do understand that the authors caution against it, but see also my points below.

p.13 l.312-13: "In this case, the generality of the parameter to interpret observations in different conditions is lost." Please explain what is meant by this statement, and how that would not be the case by separating out surface tension from the K parameter. Please give references to previous work debating this, as well as to the original work of Petters and Kreidenweis (2007) and following articles to support why this is not the proper way to interpret their presented framework.

p.13 l.318-19: "... a specific suberic acid hygroscopicity alone cannot explain the observations across a range of particle sizes and compositions" That was also found for other well-defined organic aerosol by e.g. Kristensen et al. (2014), Hansen et al. (2015), and Wittbom et al. (2018). Indications of similar relations have been seen for ambient aerosol by e.g. Hong et al. (2014,2015).

p.15 l.349-50: "...all organic material is assumed to reside in a water-free organic

shell-phase (an organic film) at the surface of the aqueous droplet." This assumption was also used by e.g. Prisle et al. (2011).

p.15 l.349-50: "...neither of the models fully captures the observed behavior at all for g and size regimes." Previous studies have made similar conclusions for other aerosol and other models, please bring these into perspective to present a broader perspective of the issue.

p.17 l.399-400: "Surface tension effects can lead to significant differences from classic, hygroscopicity mixing rule mechanisms for CCN activation." This was also seen by Kristensen et al. (2014) and Hansen et al. (2015), among others.

p.30: For example Shulman et al. (1996), Sorjamaa et al. (2004), and Prisle et al. (2008) present similar Köhler curves. Please discuss Fig. A2 in context of previous results.

The most significant novelty of this work lies in the measurements presented in Fig. 3, but these results lack a comprehensive discussion. Significantly more novel content should be added to the work before publication. For example, no explanation is given for the variation between the salt seeds and no modeling is done to provide interpretation beyond general speculation. Would it not be possible and facilitating to the discussion to make model predictions using exactly the outlined framework, for these aerosol? CCN activation experiments could be made with the systematic variation in particle size and organic fraction which this technical note specifically calls for. How are the impacts on water activity from the different salts treated in the model? Could intrinsic salt hygroscopicity/solubility explain the variation with mass fraction, and if not, why? More generally, a discussion could be added on the atmospheric/organic aerosol process relevance of suberic acid and mixtures with each of the salts.

For the model part, I commend the effort to use a more general activity model for calculating the Köhler curves. This is indeed one of the major sources of discussion related to identifying surface tension effects in Köhler theory. However, the use data

presented in Table 1 is not sufficiently justified. For example, why was the 20 C values used? What is the basis for assuming similar surface tension based on structural similarity? For other compounds, small changes in molecular length can lead to greatly enhanced surface activity. How did Riipinen et al. make the determination of the pure compound value? How is pure ammonium sulfate (a solid at 20 C) surface tension justified? How is the minimum shell thickness estimated?

What would be the impact on Köhler curves using more mixture specific input data? Can the authors rule out that any of the effects described are not caused by solution non-ideality and misrepresentation of the experimental system by use of incompatible thermodynamic data? It would strengthen the arguments significantly to use accurate reference input data if such may be obtained from a thorough literature search, or from performing new measurements for the specific systems in question. The lack of proper input data for different Köhler models is a major limitation of obtaining a comprehensive molecular framework, as called for in this technical note – this would be one of my first additions to the discussion.

The discussion about discontinuities in the activation curves is very interesting. It would be useful to see Fig. 1(B) developed for somewhat more realistic conditions. Specifically, do similar discontinuities persist in Köhler curves when concentration dependent partitioning is taken into account? What are the relation between the curves in Fig. 1(B) and A2? Why do they look different?

The present technical note draws primarily on the previous work of Ruehl et al. (2014,2016). It should be made clearer which results are summarized from there, and which are new for this work.

The presentation and arguments would benefit greatly from an independent presentation of the model framework and main assumptions used. In particular, it is not entirely clear what partitioning framework is used and how the surface composition is actually evaluated. It is not clear how mutually different the employed frameworks are, that is,

how wide a range of possible representations of the aerosol are mapped with the three different models used. As the majority of previous work uses a Gibbs-based partitioning model, would it not make sense to include such calculations in the comparison? The results in Fig. A2 could be discussed in greater depth, as could the statement "... the simplified organic film model provides the best description of these experimental data" p. 16 l.374. Why was the "best framework" not verified by e.g. measuring the droplets size at various equilibrated RH?

Specific comments.

p.3. l.66: What is meant by "vastly different"?

Fig. 2: Why are results shown for different values of Korg (dashed line) in panels A, B, and C?

Fig. 3: How are experimental error bars estimated?

Fig A2: What causes the kink in (presumably) the pure water surface tension curve in (e)?

References:

Michelle L. Shulman, Michael C. Jacobson, Robert J. Carlson, Robert E. Synovec, and Toby E. Young: Dissolution behavior and surface tension effects of organic compounds in nucleating cloud droplets, Geophys Res Lett 23, 277-280, 1996.

Sorjamaa, R., Svenningsson, B., Raatikainen, T., Henning, S., Bilde, M., and Laaksonen, A.: The role of surfactants in Köhler theory reconsidered, Atmos. Chem. Phys., 4, 2107-2117, https://doi.org/10.5194/acp-4-2107-2004, 2004.

Svenningsson, B., Rissler, J., Swietlicki, E., Mircea, M., Bilde, M., Facchini, M. C., Decesari, S., Fuzzi, S., Zhou, J., Mønster, J., and Rosenørn, T.: Hygroscopic growth and critical supersaturations for mixed aerosol particles of inorganic and organic compounds of atmospheric relevance, Atmos. Chem. Phys., 6, 1937-1952,

https://doi.org/10.5194/acp-6-1937-2006, 2006.

Petters, M. D. and Kreidenweis, S. M.: A single parameter representation of hygroscopic growth and cloud condensation nucleus activity, Atmos. Chem. Phys., 7, 1961-1971, https://doi.org/10.5194/acp-7-1961-2007, 2007.

N. L. Prisle, T. Raatikainen, R. Sorjamaa, B. Svenningsson, A. Laaksonen and M. Bilde (2008): Surfactant partitioning in cloud droplet activation: a study of C8, C10, C12 and C14 normal fatty acid sodium salts, Tellus, 60B, 416-431, DOI: 10.1111/j.1600-0889.2008.00352.x.

Prisle, N. L., Raatikainen, T., Laaksonen, A., and Bilde, M.: Surfactants in cloud droplet activation: mixed organic-inorganic particles, Atmos. Chem. Phys., 10, 5663-5683, https://doi.org/10.5194/acp-10-5663-2010, 2010.

M. Frosch, N. L. Prisle, M. Bilde, Z. Varga and G. Kiss (2011): Joint effect of organic acids and inorganic salts on cloud droplet activation, Atmospheric Chemistry and Physics, 11, 3895-3911, doi: 10.5194/acp-11-3895-2011.

T. B. Kristensen, N. L. Prisle, and M. Bilde: Cloud droplet activation of mixed model HULIS and NaCl particles: Experimental results and $\kappa$-Köhler theory, Atmospheric Research, 137, 167–175, 2014, DOI: 10.1016/j.atmosres.2013.09.017.

Hong, J., Häkkinen, S. A. K., Paramonov, M., Äijälä, M., Hakala, J., Nieminen, T., Mikkilä, J., Prisle, N. L., Kulmala, M., Riipinen, I., Bilde, M., Kerminen, V.-M., and Petäjä, T.: Hygroscopicity, CCN and volatility properties of submicron atmospheric aerosol in a boreal forest environment during the summer of 2010, Atmos. Chem. Phys., 14, 4733-4748, https://doi.org/10.5194/acp-14-4733-2014, 2014.

Hansen, A. M. K., Hong, J., Raatikainen, T., Kristensen, K., Ylisirniö, A., Virtanen, A., Petäjä, T., Glasius, M., and Prisle, N. L.: Hygroscopic properties and cloud condensation nuclei activation of limonene-derived organosulfates and their mixtures with ammonium sulfate, Atmos. Chem. Phys., 15, 14071-14089, https://doi.org/10.5194/acp-

15-14071-2015, 2015.

Hong, J., Kim, J., Nieminen, T., Duplissy, J., Ehn, M., Äijälä, M., Hao, L. Q., Nie, W., Sarnela, N., Prisle, N. L., Kulmala, M., Virtanen, A., Petäjä, T., and Kerminen, V.-M.: Relating the hygroscopic properties of submicron aerosol to both gas- and particle-phase chemical composition in a boreal forest environment, Atmos. Chem. Phys., 15, 11999-12009, https://doi.org/10.5194/acp-15-11999-2015, 2015.

Wittbom, C., Eriksson, A.C., Rissler, J. et al.   J Atmos Chem (2018). https://doi.org/10.1007/s10874-018-9380-5

---

## Referee Comment (RC2) · Anonymous Referee #2 · 18 Jan 2019

While it is not entirely clear to me why this is being framed as a "technical note," and not just a paper, I find the results to be generally interesting. There is effectively a very long introduction in which the authors lament that others continue to neglect the role that surface tension can play, followed by some interesting discussion of some new experiments. I suggest that the discussion about the new experiments is expanded.

L150: I find the terminology "schematic dependences" to be a bit unclear. It would be good to see this explained a bit further. Do these come from anywhere in particular, or are they totally made up? What sort of cases are they meant to illustrate, i.e. why these three?

L170: It is not clear to me how the statement that "This...supports assertions that surface tension does not impact activation..." follows from the previous sentence. In

the examples, the activation diameter for all three non-constant surface tension curves is lower than that obtained assuming a constant surface tension equal to that of pure water. In fact, the authors go on to say just this. So, it would seem that the activation diameter is impacted. It would be helpful if the authors could clarify their particular point here.

L182: The "discontinuity" that the authors refer to here is not entirely clear to me. Do the authors mean that each of the three variable curves shown suddenly changes relatively to assuming a constant surface tension of 30 mN/m? This is not clear from the words, as the discontinuity does not occur "at its minimum surface tension" but instead when there is a (in the model used here) discontinuous increase in the minimum surface tension.

L235: It would be helpful if the "relatively simple , phase composition- and morphology-specific surface tension model" were described here, in brief. If I understand correctly, a linear mixing rule is used.

L235: It would also be helpful if the authors were to clarify whether the model predicted the surface activation effect completely independent of the observations, or whether some amount of model tuning was included (perhaps associated with exactly what species were chosen to represent the organic phase).

L244: I suggest that the authors modify the sentence ending "process" to say "process, for these systems." It is not clear that this is the case for all systems. If it were, we might expect to see a lack of closure more generally in ambient measurements.

L251: What is stated does not appear to be the case for the example shown in Fig. S3.7.4 in the Ovadnevaite paper.

L305: I do not find it clear what the authors mean when they state that the compressed-film model "more closely predict[s] the observations." More closely than what? The simple model assuming a constant kappa_org = 0.35 seems to do quite well. I do get

that the authors are working to say that kappa_org = 0.35 is not the correct answer and thus the good agreement is not for the right reason. Yet this discussion requires clarification.

L315: The evidence that the apparent (unrealistic) kappa_org is particle size dependent is very compelling.

L316: Is the compressed film model here simply inadequate, even with determination of new parameters? Or can it work if it were retuned to this dataset? If the latter, this would give some indication of how uncertainty in the derived parameters for the model impacts subsequent calculations.

L328: This is only true for the smallest particles.

L330: I am not seeing entirely what information is lost by choice of procedure or data representation. Both ways of representing the data seem to drive one to similar conclusions: for a given representation the assumption of a constant kappa_org can allow for model-measurement agreement, but the kappa_org values are not necessarily realistic. I do not understand why the authors couldn't similarly apply the models shown in Fig. 2A/B to the data in Fig. 2C? There need not be loss of information.

L348: It is not clear to me how these are AIOMFAC calculations any longer, if the organic and salt are forced into separate phases. Does one even need AIOMFAC to do this?

L361: How is it that the essentially made up model (assuming a complete surface film) does better than the model that explicitly accounts for the system thermodynamics? Is this just saying that the thermodynamic model is wrong (e.g. bad parameterization or parameterization used outside the RH range where it is valid)? Or is there perhaps some kinetic effect that must be accounted for? The discussion that comes a few sentences later seems to suggest perhaps the former. But, some additional and more explicit discussion is necessary, rather than a statement (line 375) that this finding is

"intriguing".

Minor Comments:

Fig. 1: It would be good to change from the red/green color scheme to one that is more friendly to color blind people.
* * *

---

## Author Comment (AC1) · 12 Feb 2019

Response to Reviewer 1:

*The technical note summarizes a set of interesting points to be clarified regarding the role of surface active organic aerosol in cloud activation. In light of several recent works, including those of the authors, this topic is very timely and their call for a discussion on the topic is commendable. While it is useful on occasion to summarize the knowledge and open questions in the field, my main concern is that the note brings very little new to the table. Most or all of the points have been made previously, but little references or perspective is given to the substantial body of work already existing on the subject.*

We thank the reviewer for his or her comments. As a technical note, the intention of this work is to clarify a key technical point that is often obscured – namely that the evolution of surface tension, in addition to its absolute value at any given droplet size, is an important factor in regulating the activation of CCN. We acknowledge that the theory behind this is established, but it is only in recent experimental efforts that direct observations of modification to Kohler curves have been possible. In these works, the evolving surface tension is shown to change the size of the activating particles, with activation occurring at much larger wet diameters. However, the effect on the activation efficiency (through $S_{crit}$) is more subtle and can easily be missed when performing typical CCNC experiments. Thus, we aim to clarify why surface tension affects Kohler curves in this manner, and to convey the situations where surface tension effects might be notable and detectable using standard CCN instrumentation. We do not seek to provide a complete or exhaustive review of the literature in the very broad science of CCN activation, instead our focus is on two main points:

      1. The evolution of surface tension is the main driver for changes in CCN activation relative to assumptions of fixed surface tension.

      2. Current models struggle to explain and predict experimentally observed data across a wide range of parameter space (size, organic fraction and inorganic composition), but experiments can be performed that clearly reveal surface effects which are needed to help to drive the development of new theories and practical, predictive models of CCN growth and activation by the community.

*p.2 l. 36-38: "In this note, we focus on the role of surface tension, and discuss the limitations of current approximations in light of recently published works that reveal how it is primarily the evolution of surface tension that impacts the activation process." Only two papers are discussed in detail, the works of Ruehl et al. (2016) and Ovadnevaite et al. (2017). The discussion is not sufficiently substantiated independently of these works.*

To our knowledge, it is only in recent papers that experimental observations have been able to corroborate the assertion that surface tension influences hygroscopic growth and CCN activation in organic and mixed inorganic-organic aerosol (Forestieri et al., 2018; Ruehl et al., 2016). These recent studies have done this via direct measurements of droplet size at known saturation ratios (i.e Kohler curves) that show the shape of the curve as activation is approached. While much previous work has discussed the role of surface tension, as noted in our manuscript, there remain only very few studies that directly measure the effects. We wish to emphasize that now the goal is

to find a model description to best describe the data over all observable space, based on established physical parameters, rather than fitting an empirical or semi-empirical model to individual datasets. This is challenging, as in many cases surface tension effects are obscured by the way data is presented or analyzed. Indeed, we show that the common practice of showing $S_{crit}$ versus $D_{dry}$ is not effective at highlighting differences that may be attributed to a variable surface tension, due primarily to the larger role that the particle size has in the Kohler equation.

***p.3 l.72-75: Effects of inorganic co-solutes is discussed by e.g. Svenningsson et al. (2006), Prisle et al. (2010,2011), Frosch et al. (2011), Kristensen et al. (2014), and Hansen et al. (2015). Effects of dissolution is discussed by e.g. Shulman et al. (1996) and Bilde and Svenningsson (2004).***

We thank the reviewer for their recommended additional literature to include in our work, and these have been added to this discussion, indicated below. We acknowledge that there are significant bodies of work outside of those cited in our Technical Note, though we have chosen to avoid a comprehensive review of the field in light of the focus of the manuscript.

"Furthermore, surface partitioning may be influenced by non-surface active components in the system, such as the presence of inorganic material and co-solutes (Asa-Awuku et al., 2008; Boyer et al., 2016; Boyer and Dutcher, 2017; Frosch et al., 2011; Petters and Petters, 2016; Prisle et al., 2011; Svenningsson et al., 2006; Wang et al., 2014). Other factors that have been shown to influence the shape of Köhler curves are: (1) solute dissolution, encompassing both water-solubility and solubility kinetics (Asa-Awuku and Nenes, 2007; Bilde and Svenningsson, 2017; McFiggans et al., 2006; Petters and Kreidenweis, 2008; Shulman et al., 1996), (2) liquid–liquid phase separation (i.e. limited liquid–liquid solubility) (Rastak et al., 2017; Renbaum-Wolff et al., 2016), and (3) the dynamic condensation (or gas–particle partitioning) of organic vapors (Topping et al., 2013; Topping and McFiggans, 2012) (Topping et al., 2013; Topping and McFiggans, 2012).

***p.4. l.81: ". . .there are many observations that cannot be explained in such simple terms". Please be specific, which observations cannot be explained and what is the argument that they are unexplained.***

The studies cited apply κ-Köhler theory without fully considering the potential influence of surface tension. Collins et al., 2016 note that the method of sea-spray production from bubble bursting likely leads to aerosol more highly concentrated in surface active species. Good et al., 2010 suggest surface tension as a potential source of the deviation in their CCN concentrations. Yakobi-Hancock et al., 2014 briefly discuss data at high organic fraction that shows a much higher $\kappa_{org}$ than would be expected (albeit with some large uncertainty). These papers demonstrate the conditions where surface tension effects are likely to be significant – the presence of organic molecules with rather low water-solubility present in relatively high concentrations in small particles, as concluded in our manuscript. Thus, these observations cannot be fully explained without considering surface effects. We have modified our language to better reflect the intended meaning:

"While measured cloud droplet number concentrations in the atmosphere have been explained in several cases with simple parameterizations that neglect dynamic surface effects (Nguyen et al., 2017; Petters et al., 2016), there are many observations that are not **fully** explained in such simple terms **and in those cases a substantial population of CCN may exhibit behavior characteristic**

**of surface effects** (Collins et al., 2016; Good et al., 2010; Ovadnevaite et al., 2011; Yakobi-Hancock et al., 2014)"

*p. 5-6 and Fig. 1(A): A substantial part of the manuscript is devoted to explaining basic Köhler theory, as done by many authors previously, including basic textbook material. The presentation could benefit from a similarly detailed presentation of the surface tension and partitioning models used.*

We carefully state the factors that go into the Köhler equations in order to be thorough and precise in our message. Without fully describing the basic equations, it can be hard to see how water activity and the Kelvin effect affect the results, and how these must be considered separately. A detailed description of the partitioning models of Ruehl et al. (2016) and Ovadnevaite et al. (2017) can be found in their respective manuscripts. Additional information about the AIOMFAC-based models was added to the manuscript on line 236 (line numbers of original manuscript); see our reply to referee #2. However, it is clear that a complete, accurate and predictive description of CCN activation applicable in the general case is not yet available. Rather, our intention is to promote additional laboratory studies and analysis to identify more cases where surface tension has real and measurable effects, in an effort to gain the insight required for producing a predictive theory that captures the full complexity of CCN activation. Thus, we take a step back and focus on the fundamentals of Köhler theory that necessitate the development of the partitioning / surface tension models.

*p.7 l. 171-72: "This also seemingly supports previous assertions that surface tension does not impact activation, since it is generally argued that at the point of activation the droplet is sufficiently dilute and essentially exhibits a surface tension like pure water." Please reference a sufficient amount of such previous assertions to support the claim of a general argument.*

Earlier in the text, we discuss the references that point to this assertion (lines 44+) and we have amended the text to refer the reader back to this section:

"Traditionally, however, surface tension has been reduced to a fixed term in the Köhler equation (Abdul-Razzak and Ghan, 2000; Facchini et al., 2000; Petters and Kreidenweis, 2007) and is usually given a temperature-independent value equal to that of pure water at 25 °C. This is because for any decrease in surface tension due to bulk–surface partitioning and surface adsorption, it is assumed that there is an increase in the solution water activity because adsorbed material, previously acting as a hygroscopic solute, is removed from the droplet (bulk) solution (Fuentes et al., 2011; Prisle et al., 2008; Sorjamaa et al., 2004). Thus, the effects approximately cancel out in the calculation of a droplet's equilibrium saturation ratio via the Köhler equation and so are often neglected. Furthermore, it has been shown in some cases that there is insufficient material in a droplet at the sizes approaching activation to sustain a surface tension depression (Asa-Awuku et al., 2009; Prisle et al., 2010)."

*p.8 l.180: ". . . knowing the surface tension only at the point of activation is insufficient to determine the critical properties at activation. . ." Please provide references.*

We have rephrased this statement to clarify our meaning, as this reflects an important point. The text here now reads:

"Clearly, an evolving surface tension prior to activation can matter in such systems and, consequently, knowing the surface tension only at the point of activation is in a general case insufficient for determining the critical properties at activation (absent any droplet size measurement), because the position of the maximum in the Köhler curve and, thus, the droplet size at activation, will depend on the trajectory of surface tension evolution. Moreover, knowing the surface tension at the activation point only, may not allow for an accurate prediction of whether a droplet of given dry diameter will activate at a given environmental supersaturation (compare the green curve with the *iso-σ* curve of 72 mN m$^{-1}$ in Fig. 1B, both having the same surface tension at their points of CCN activation, yet different critical supersaturations and wet diameters). Similar conclusions have been drawn previously (e.g. Prisle et al., 2008), although the measurements by Ruehl et al. (Ruehl et al., 2016) were the first to verify this experimentally."

***p.11 l.258-64: "This predicted size effect indicates that it is not generally valid to assume that all activating CCN will have a surface tension equivalent to or close to that of pure water –nor is it appropriate to use a single measurement of the surface tension of a multicomponent CCN of known dry composition at its activation size (only) to determine its Köhler curve. Furthermore, these model predictions also suggest that measurements of the surface tension of larger CCN particles (e.g. > 150 nm dry diameter) may not allow for conclusions about the surface tension of much smaller CCN, e.g. of 50 nm dry diameter." These points have certainly been made by other studies than the one specifically described, including those mentioned above, and Wittbom et al. (2018). A broader perspective should be provided.***

Studies that do not explicit account for surface partitioning, such as Wittbom et al., indicate there is a size-dependence to their values of κ, and implying to us that there might be surface tension effects at play. The studies of e.g. Prisle et al. (2008), Prisle et al. (2010) and Sorjamaa et al. (2004), now cited in the text at this point, discuss size-dependent surface tension but do not appear to show clear deviations from an assumption that surface tension is size-independent. Indeed, Sorjamaa et al. conclude the opposite of the observations reported here – that surface effects become more significant with larger particles. As we show in Figure 2, we note the opposite, that smaller particles of the same composition are more CCN active and thus suggests greater surface tension effects. We have now cited additional papers in this context that discuss size dependencies relating to surface partitioning.

**p.12 l. 292-92: ". . .in contrast to other studies that allow both coated particle size and organic volume fraction to vary simultaneously." Although in many studies particles are produced by other methods than coating (e.g. nebulization), again many other works have indeed recognized the importance of independently varying the particle size and organic fraction, for studying the effects of organic surface activity and mixing effects, including those listed above.**

We acknowledge that other studies have performed measurements in this manner and have now included references to Prisle et al. (2010) and Wittbom et al. (2018). Many studies do however

only show data where f_org is varied in conjunction with D_dry, which we argue conceals subtle surface effects that might influence the system.

*p.13 l.310-12: "One could ignore the physical meaning of K and simply use it as an all-encompassing parameter to describe activation efficiency." This was already done by several previous studies, e.g. Kristensen et al. (2014) and Hansen et al. (2015). I do understand that the authors caution against it, but see also my points below. p.13 l.312-13: "In this case, the generality of the parameter to interpret observations in different conditions is lost." Please explain what is meant by this statement, and how that would not be the case by separating out surface tension from the K parameter. Please give references to previous work debating this, as well as to the original work of Petters and Kreidenweis (2007) and following articles to support why this is not the proper way to interpret their presented framework.*

When comparing the formulation of κ-Köhler theory by Petters and Kreidenweis (2007) to the complete Köhler theory, the κ parameter is equal to the ratio of the molar volume of water to the molar volume of a non-electrolytic solute for an ideal solution – therefore representing a simplified model for water activity. Thus, deviations from this exact value can be attributed to non-ideality. Water activity, however, is a function of the bulk concentration, and thus should be decoupled from surface effects (and be size invariant). By using κ to account for both water activity and surface tension, a size dependence is introduced due to bulk–surface partitioning and surface-to-volume ratio, and thus its generality is lost. In the general Köhler equation, water activity is a factor distinct from the Kelvin factor and therefore one distinguishes between surface tension effects and bulk solution non-ideality (which relates to hygroscopicity also at RH < 100 %). By using κ as an adjustable fit-parameter, one may be able to fit experimental data, but by doing so, the retrieved κ will be system-specific and size-specific if the actual surface tension of the droplets varied with size. Such system- and size-specific fitted models will not serve the purpose of establishing a predictive theory and model. This is why we caution against it.

*p.13 l.318-19: ". . . a specific suberic acid hygroscopicity alone cannot explain the observations across a range of particle sizes and compositions" That was also found for other well-defined organic aerosol by e.g. Kristensen et al. (2014), Hansen et al. (2015), and Wittbom et al. (2018). Indications of similar relations have been seen for ambient aerosol by e.g. Hong et al. (2014,2015).*

The mentioned studies report agreement when looking at critical supersaturation vs $D_{dry}$ ($D_{coated}$ in our manuscript). Indeed, Kristensen et al. conclude that a fixed surface tension of water and a fitted κ value for the organic fraction gives best agreement. Hansen et al. are not definitive in their conclusions of a size dependent κ, although they do indicate that smaller particles exhibit a greater surface tension reduction, and the focus of Wittbom et al. is on solubility, which may show different size dependencies than surface tension and is thus not directly relevant for the discussion here. The data and choice of coordinates we report in Figure 2 is the most clear demonstration that a single value of κ cannot reproduce the data for this system across different sizes. Decoupling size from composition allows for a more rigorous assessment of the effectiveness of a model in account for each parameter. $D_{dry}$ is not a sensitive metric for observing deviations due to surface tension because both size and composition vary, and their effect on the extent of surface tension

effects is opposite. That is to say, as $D_{dry}$ gets smaller (likely to increase role of surface tension), the organic fraction decreases (likely to decrease role of surface tension). Thus, the influence of surface tension is masked by the manner in which the data is presented.

***p.15 l.349-50: ". . .all organic material is assumed to reside in a water-free organic shell-phase (an organic film) at the surface of the aqueous droplet." This assumption was also used by e.g. Prisle et al. (2011).***

This assumption, as we discuss, is the most simple to apply, and yields the best results for the sample case discussed. Given that it is not well-supported by the applied partitioning theory for the case of suberic acid, it is interesting that this is the case and suggests we have a long way to go to quantitatively understand what is happening during the CCN activation process. In Prisle et al. (2011), this assumption is applied for the case of "classic" surfactants for which conceptually such an assumption might seem more appropriate. In our case, this assumption still yields good results, despite the molecules not being true surfactants. In general, this organic film assumption represents a limiting case, and in reality a balance between surface partitioning and water activity suppression should explain the results. We have added the following sentence to acknowledge these previous studies:

"This observation is consistent with the results of (Prisle et al., 2011) who applied a similar simple model to droplets containing ionic surfactants and hints at a significant suppression of surface tension by suberic acid, which is likely highly enriched at the droplet surface"

***p.15 l.349-50: ". . .neither of the models fully captures the observed behavior at all forg and size regimes." Previous studies have made similar conclusions for other aerosol and other models, please bring these into perspective to present a broader perspective of the issue.***

[The referee refers to line 359]  The discussion in this section of the manuscript concerns the system and models shown in Fig. 2. To discuss previous literature which makes similar conclusions for other aerosol systems – yet not presented in terms of organic volume fraction at constant size cases – does, in our opinion, not add to the purpose of the discussion at this point in the article.

***p.17 l.399-400: "Surface tension effects can lead to significant differences from classic, hygroscopicity mixing rule mechanisms for CCN activation." This was also seen by Kristensen et al. (2014) and Hansen et al. (2015), among others.***

We have added additional citations here to refer readers to the works the reviewer suggests.

***p.30: For example Shulman et al. (1996), Sorjamaa et al. (2004), and Prisle et al. (2008) present similar Köhler curves. Please discuss Fig. A2 in context of previous results.***

This type of Köhler curve is produced when additional factors that determine the water activity or surface tension vary with the changing particle size. Solubility limitations and surface tension suppression both produce plots that may exhibit discontinuities. The point of Fig. A1 and associated text in the manuscript is to illustrate the different outcomes of the employed AIOMFAC-based model variants for a specific case of the aqueous suberic acid + ammonium

sulfate system including accounting for liquid–liquid solubility limitations. We do agree with the referee that Köhler curves similar in shape have been produced by other models for other systems in the past, including in the work by Shulman et al. (1996) but we note also that the shapes of such variable-σ Köhler curves were only recently verified experimentally.

*The most significant novelty of this work lies in the measurements presented in Fig. 3, but these results lack a comprehensive discussion.*

These results were included to highlight the other factors that influence observations and demonstrate that co-solute effects are important to consider. These observations must also be accounted for in any predictive overall model, and are thus included to illustrate the need for a comprehensive theory applicable over the relevant parameter space.

*Significantly more novel content should be added to the work before publication. For example, no explanation is given for the variation between the salt seeds and no modeling is done to provide interpretation beyond general speculation. Would it not be possible and facilitating to the discussion to make model predictions using exactly the outlined framework, for these aerosol? CCN activation experiments could be made with the systematic variation in particle size and organic fraction which this technical note specifically calls for. How are the impacts on water activity from the different salts treated in the model? Could intrinsic salt hygroscopicity/solubility explain the variation with mass fraction, and if not, why? More generally, a discussion could be added on the atmospheric/organic aerosol process relevance of suberic acid and mixtures with each of the salts.*

As a technical note focused on conveying a single point, the role of evolving surface tension on CCN activation, a broader discussion is beyond the scope of this note. Moreover, we emphasize that the goal for showing the data of Fig. 3 is not in fitting a specific (simple or sophisticated) CCN activation model to the data presented in Fig. 3; rather, the point is made that additional complexity is observed with respect to organic–inorganic interactions stemming from different types of solute species. We agree that these data are particularly interesting, and hope further studies will be able to learn from what we present here and offer more tangible theoretical frameworks that allow the relevant physical parameters to be accurately included.

*For the model part, I commend the effort to use a more general activity model for calculating the Köhler curves. This is indeed one of the major sources of discussion related to identifying surface tension effects in Köhler theory. However, the use data presented in Table 1 is not sufficiently justified. For example, why was the 20 C values used? What is the basis for assuming similar surface tension based on structural similarity? For other compounds, small changes in molecular length can lead to greatly enhanced surface activity. How did Riipinen et al. make the determination of the pure compound value? How is pure ammonium sulfate (a solid at 20 C) surface tension justified? How is the minimum shell thickness estimated? .*

Adipic acid and suberic acid are the closest in size of the dicarboxylic acid series, and both contain even numbers of carbons. It is indeed true that smaller changes can result in large differences of surface tension, but this is typical seen when going from even to odd, and that adjacent even or odd compounds are more similar (Ruehl and Wilson, 2014). The tabulated value for suberic acid

is that estimated for the pure liquid (subcooled) component; its value is likely similar to that of pure liquid (subcooled) adipic acid and other dicarboxylic acids. Adipic acid's pure value was determined theoretically using the Macleod-Sudgen method, as described in Riipinen et al. A pure component surface tension is a physicochemical property that is distinct from a molecule's surface affinity in a solution droplet (however, when such a substance contributes a substantial area fraction of the surface, it's surface tension value will affect that of the droplet). A temperature of 20˚C was used because this was the ambient temperature in which the experiments were conducted. The ammonium sulfate was not considered to influence surface tension at dilute aqueous conditions, as noted in the footnote to Table 1. We have amended the text in the table to state "surface tension of aqueous AS (at T)". The minimum shell thickness is one of the assumptions applied in the predictive AIOMFAC-based model and is established based on the approximate size of the molecules and related monolayer thickness.

*What would be the impact on Köhler curves using more mixture specific input data? Can the authors rule out that any of the effects described are not caused by solution non-ideality and misrepresentation of the experimental system by use of incompatible thermodynamic data? It would strengthen the arguments significantly to use accurate reference input data if such may be obtained from a thorough literature search, or from performing new measurements for the specific systems in question. The lack of proper input data for different Köhler models is a major limitation of obtaining a comprehensive molecular framework, as called for in this technical note – this would be one of my first additions to the discussion.*

We agree with the reviewer that any model should use input data which are directly measured rather than fit to the experimental data in question. We are not sure what misrepresentation the reviewer might be referring to – the main conclusion of the paper, as supported by the experimental results, is that we do not have a correct model, and perhaps lack the correct inputs or physicochemical details to fully understand what is happening. However, we do consider bulk phase non-ideality in the models explicitly, including organic–inorganic interactions in the AIOMFAC model, aspects often ignored in other models. We are confident that the thermodynamic data used for model inputs are correct (or reasonable estimates where direct measurements are lacking). The application of a spectrum of models in this work is to demonstrate that surface effects can reasonably explain and constrain the observations, but that a robust, accurate and fully predictive model for such coupled hygroscopicity and surface tension effects is yet to be developed.

*The discussion about discontinuities in the activation curves is very interesting. It would be useful to see Fig. 1(B) developed for somewhat more realistic conditions. Specifically, do similar discontinuities persist in Köhler curves when concentration dependent partitioning is taken into account? What are the relation between the curves in Fig. 1(B) and A2? Why do they look different?*

Figure 1(B) is a schematic using exaggerated surface tension dependencies to highlight the differences that may be encountered based on different surface tension evolutions. Those shown in A2 result from the AIOMFAC-based model variants for the suberic acid + AS system, as described in the manuscript. Not all possible schematic surface tension evolutions are exhibited by

a single system. Discontinuities in Köhler curves, similar to the schematic in Fig. 1B, have been observed e.g. by Ruehl et al. (2016) for droplets containing ammonium sulfate and various dicarboxylic acids. Those observations and discontinuities were reproduced by a (fitted) compressed film model of CCN activation.

***The present technical note draws primarily on the previous work of Ruehl et al. (2014,2016). It should be made clearer which results are summarized from there, and which are new for this work. The presentation and arguments would benefit greatly from an independent presentation of the model framework and main assumptions used. In particular, it is not entirely clear what partitioning framework is used and how the surface composition is actually evaluated. It is not clear how mutually different the employed frameworks are, that is, how wide a range of possible representations of the aerosol are mapped with the three different models used. As the majority of previous work uses a Gibbs-based partitioning model, would it not make sense to include such calculations in the comparison? The results in Fig. A2 could be discussed in greater depth, as could the statement ". . . the simplified organic film model provides the best description of these experimental data" p. 16 l.374. Why was the "best framework" not verified by e.g. measuring the droplets size at various equilibrated RH?***

The main intention of this paper is not to present a "best" model in the sense of a recommendation to the community. The main goal is to show that current models are insufficient in capturing the details and the dynamics across a range of parameter space relevant to CCN activation. While different from classical Szyszkowski-Langmuir isotherm model for bulk-surface partitioning and surface tension, the compressed film model by Ruehl et al. (2016) employed in this work is also a model that includes a thermodynamically consistent isotherm model for relating surface composition to bulk concentration. A more detailed description of the models is therefore unwarranted.

**References**

Collins, D. B., Bertram, T. H., Sultana, C. M., Lee, C., Axson, J. L. and Prather, K. A.: Phytoplankton blooms weakly influence the cloud forming ability of sea spray aerosol, Geophys Res Lett, 43(18), 9975--9983, doi:10.1002/2016gl069922 , 2016.

Forestieri, S. D., Staudt, S. M., Kuborn, T. M., Faber, K., Ruehl, C. R., Bertram, T. H. and Cappa, C. D.: Establishing the impact of model surfactants on cloud condensation nuclei activity of sea spray aerosol mimics, Atmos Chem Phys, 18(15), 10985–11005, doi:10.5194/acp-18-10985-2018 , 2018.

Good, N., Topping, D., Allan, J., Flynn, M., Fuentes, E., Irwin, M., Williams, P., Coe, H. and McFiggans, G.: Consistency between parameterisations of aerosol hygroscopicity and CCN activity during the RHaMBLe discovery cruise, Atmos Chem Phys, 10(7), 3189--3203, doi:10.5194/acp-10-3189-2010 , 2010.

Prisle, Nø. L., Raatikainen, T., Sorjamaa, Svenningsson, B., Laaksonen, A. and Bilde, M.: Surfactant partitioning in cloud droplet activation: a study of C8, C10, C12 and C14 normal fatty acid sodium salts, Tellus B, 60(3), 416–431, doi:10.1111/j.1600-0889.2008.00352.x , 2008.

Ruehl, C. R. and Wilson, K. R.: Surface Organic Monolayers Control the Hygroscopic Growth of Submicrometer Particles at High Relative Humidity, J Phys Chem, 118(22), 3952–3966, doi:10.1021/jp502844g , 2014.

Ruehl, C. R., Davies, J. F. and Wilson, K. R.: An interfacial mechanism for cloud droplet formation on organic aerosols, Science, 351(6280), 1447–1450, doi:10.1126/science.aad4889 , 2016.

Yakobi-Hancock, J., Ladino, L., Bertram, A., Huffman, J., Jones, K., Leaitch, W., Mason, R., Schiller, C., Toom-Sauntry, D., Wong, J. and Abbatt, J.: CCN activity of size-selected aerosol at a Pacific coastal location, Atmos Chem Phys, 14(22), 12307–12317, doi:10.5194/acp-14-12307-2014 , 2014.

---

## Author Comment (AC2) · 12 Feb 2019

Response to Reviewer 2:

*While it is not entirely clear to me why this is being framed as a "technical note," and not just a paper, I find the results to be generally interesting. There is effectively a very long introduction in which the authors lament that others continue to neglect the role that surface tension can play, followed by some interesting discussion of some new experiments. I suggest that the discussion about the new experiments is expanded.*

We thank the reviewer for his/her time in reviewing our manuscript. We present this work as a technical note in order to convey the key point to the work – that it is the evolution of surface tension, in addition to the absolute value at any given droplet size, that is important in regulating the activation of CCN. The introduction serves to ensure the reader has sufficient context in order to appropriately apply and considerer surface tension effects when assessing CCN data.

*L150: I find the terminology "schematic dependences" to be a bit unclear. It would be good to see this explained a bit further. Do these come from anywhere in particular, or are they totally made up? What sort of cases are they meant to illustrate, i.e. why these three?*

These are schematic in the sense that they do not conform to the behavior of any particular solute/solvent system. However, they represent the types of dependencies that are encountered and the resulting implications on the Kohler curves. The text now reads:

"Distinct schematic dependences of the surface tension on the droplet size, representative of the types of dependence that might be encountered in real aerosol, are imposed for the purpose of illustration, shown in the lower panel of Figure 1B"

*L170: It is not clear to me how the statement that "This. . .supports assertions that surface tension does not impact activation. . ." follows from the previous sentence. In the examples, the activation diameter for all three non-constant surface tension curves is lower than that obtained assuming a constant surface tension equal to that of pure water. In fact, the authors go on to say just this. So, it would seem that the activation diameter is impacted. It would be helpful if the authors could clarify their particular point here.*

Figure 1B shows the opposite of what is being asserted by the reviewer – each of the curves associated with a variable surface tension exhibits an activation diameter that is larger (not lower) than the constant surface tension curve. While surface tension at the point of activation may be equal to that of pure water, the simulations here indicate that the point of activation is modified by the evolution of the surface tension. The point in the mentioned sentence on L170 is that in most CCN measurements, only the activation supersaturation is measured for a given dry size, but not the activation size nor the surface tension. Thus, one might assume that the surface tension at the activation point is equal to that of pure water, which may frequently be the case. However, as the examples in Fig. 1B show, assuming that the surface tension is constantly that of pure water prior to reaching those dilute conditions and using a determined hygroscopicity parameter to predict the Köhler curve and CCN activation point would lead to a substantially different critical supersaturation and activation diameter (that of the peak in the 72 mN/m Köhler curve). Or in case of a measurement of critical supersaturation in order to determine a particle's true hygroscopicity

parameter at CCN activation, an incorrect parameter would be determined (one that would accidentally combine surface tension evolution effects and actual hygroscopicity in a non-trivial and incorrect way).

*L182: The "discontinuity" that the authors refer to here is not entirely clear to me. Do the authors mean that each of the three variable curves shown suddenly changes relatively to assuming a constant surface tension of 30 mN/m? This is not clear from the words, as the discontinuity does not occur "at its minimum surface tension" but instead when there is a (in the model used here) discontinuous increase in the minimum surface tension.*

The term "discontinuity" was used because the gradient of the Kohler curve changes discontinuously when realistic surface tension profiles are included. To better reflect the changes in the function itself, we now refer to these as abrupt changes in the text. The "second discontinuity" comments have been omitted in the revised manuscript as they distract from the more important facets of the discussion.

*L235: It would be helpful if the "relatively simple, phase composition- and morphology specific surface tension model" were described here, in brief. If I understand correctly, a linear mixing rule is used.*

At this line in the manuscript, we now add the following brief description of the AIOMFAC-based model and mixing rules for surface tension used:

"A detailed description of this AIOMFAC-based model, its variants and sensitivities to model parameters and assumptions is given in the supplementary information of Ovadnevaite et al (2017). Briefly, the equilibrium gas–particle and liquid–liquid partitioning model is used to predict the phases and their compositions for a bulk mixture "particle", here of known dry composition, at given RH and temperature. For other applications with given total gas + particle input concentrations, the equilibrium condensed-phase concentrations can be computed as a function of RH (accounting for partitioning of semivolatiles). Assuming spherical particles of a certain dry diameter with a core-shell morphology of liquid phases in the case of LLPS, density information from all constituents is used to compute volume contributions and the size of the particle at elevated RH. In addition, the surface tension of each individual liquid phase is computed as a volume-fraction-weighted mean of the pure-component surface tension values. In the case of LLPS, the surface coverage of the organic-rich shell phase is evaluated by considering that it must be greater than or equal to a minimum film thickness (monolayer as a lower limit); this determines whether complete or partial surface coverage applies for a certain wet diameter. The effective surface tension of the whole particle is then computed as the surface-area-weighted mean of the surface tensions of contributing phases. This way the surface tension evolves in a physically reasonable manner as a droplet grows, including the possibility for abrupt transitions from a low surface tension, established due to full organic droplet coverage under LLPS, to partial organic film coverage after monolayer film break-up, and further to complete dissolution of organics in the aqueous inorganic-rich phase (single aqueous phase). This equilibrium model is here referred to as AIOMFAC-EQUIL. We also employed two AIOMFAC-CLLPS model variants, in which the organic constituents are assumed to reside constantly in a separate phase from ammonium

sulfate (complete LLPS, an organic film), either with or without water present in that phase, discussed in Section 4."

**L235: It would also be helpful if the authors were to clarify whether the model predicted the surface activation effect completely independent of the observations, or whether some amount of model tuning was included (perhaps associated with exactly what species were chosen to represent the organic phase).**

In the study by Ovadnevaite et al. (2017) a predictive model based on AIOMFAC was employed. The predicted surface tension evolution by that model is a result of the calculations when liquid–liquid phase separation is included; no model tuning was performed concerning the model physics. However, the reviewer is correct that the selection of surrogate species, used to represent a particular "ambient" organic-inorganic aerosol system, will affect the predictions. In the case of the Ovadnevaite et al. study, the selection of surrogate species and their concentrations were constrained by available observations on the chemical composition of the aerosols. Obviously, no molecular-level composition information was available and as such, a range of options may be considered. Hence, the choice of system components and compositions may affect predicted CCN properties, however, beyond that, the surface tension and particle size as a function of water activity was purely a result of the computations. Extensive results and sensitivity calculations on composition and physical parameters are documented in the SI of Ovadnevaite et al. (2017) and not of relevance for the discussion in this technical note.

Concerning the modelling results described in this work, the organic to salt ratio, hence dry composition, is known and no tuning of input parameters was performed. Instead, we rely on physical parameters established independently of this work to assign feasible input parameters and compare with our experimental observations. Fitting a model to these data would not yield insight into whether our understanding of several key processes is complete. Indeed, our conclusions are that we do not have an accurate model description of the process over a range in dry particle sizes, and so we are still some way off from a predictive framework.

*L244: I suggest that the authors modify the sentence ending "process" to say "process, for these systems." It is not clear that this is the case for all systems. If it were, we might expect to see a lack of closure more generally in ambient measurements.*

We have amended the manuscript to reflect this clause.

*L251: What is stated does not appear to be the case for the example shown in Fig. S3.7.4 in the Ovadnevaite paper.*

The figure panels corresponding to the equilibrium AIOMFAC model with LLPS in that study show activation occurring at the point when surface tension returns to the value for pure water (Fig. S3.7.4a by Ovadnevaite et al.). This is in contrast, for example, to the prediction by the same model for the same system and conditions by dry particles of 41 nm diameter, which is why we point to a size dependence when the surface tension at the activation point is discussed. Other models, including the organic film variant, predict a reduced surface tension at the activation point

even for larger dry diameter, yet also that variant points to a size dependence of the surface tension at activation.

*L305: I do not find it clear what the authors mean when they state that the compressed film model "more closely predict[s] the observations." More closely than what? The simple model assuming a constant kappa_org = 0.35 seems to do quite well. I do get that the authors are working to say that kappa_org = 0.35 is not the correct answer and thus the good agreement is not for the right reason. Yet this discussion requires clarification.*

We agree this is confusing, and that statement has been removed, replaced with:

"The compressed film model decouples the water activity component from the surface tension component, and thus should better represent the physical processes at work. However, the agreement for all these models breaks down further when looking at different sized particles."

as a concluding remark prior to our discussion of size dependencies.

*L316: Is the compressed film model here simply inadequate, even with determination of new parameters? Or can it work if it were returned to this dataset? If the latter, this would give some indication of how uncertainty in the derived parameters for the model impacts subsequent calculations.*

In the current compressed film model framework, no combination of parameters was able to reproduce the trends for all size regimes considered. Some imposed additional size dependencies on the input parameters might produce good agreement, but at that point the model becomes a multivariable fit rather than a predictive physical framework.

*L328: This is only true for the smallest particles.*

This is stated on line 327 referring to the 40 nm particles.

*L330: I am not seeing entirely what information is lost by choice of procedure or data representation. Both ways of representing the data seem to drive one to similar conclusions: for a given representation the assumption of a constant kappa_org can allow for model-measurement agreement, but the kappa_org values are not necessarily realistic. I do not understand why the authors couldn't similarly apply the models shown in Fig. 2A/B to the data in Fig. 2C? There need not be loss of information.*

We are trying to show in these figures that by including size and composition in a single axis (as in the case of Figure 2C with $D_{coated}$), it becomes much harder to discern whether the model is doing a good job or not. The smallest size particles, which show the biggest surface effects when predominantly organic in nature, are at low organic volume fractions in this representation (fixed inorganic seed size), and the larger particles that do contain more organic show less surface sensitivity. Hence, for the same data, it might be reasonably concluded that a fixed value of kappa_org results in good agreement across the dataset, when in reality it is not fully capturing all the dependencies. There is no true loss of data, and we have amended our comments in the manuscript to better reflect our meaning.

"Thus, the choice of experimental procedure or data presentation can impact the interpretation of experimental observations and we caution care when presenting such data. When composition and particle size are coupled in $D_{coated}$, the sensitivity to surface tension effects is diminished, as smaller particles will contain a lower volume fraction of organic material"

**L348: It is not clear to me how these are AIOMFAC calculations any longer, if the organic and salt are forced into separate phases. Does one even need AIOMFAC to do this?**

The non-ideal mixing among the compounds within each phase is still treated with AIOMFAC, necessary to determine the water content as a function of RH (of the inorganic core phase or of both phases, depending on the model variant). The AIOMFAC-CLLPS (w/ org film) model is a simplified implementation of a possible, yet extreme case of phase partitioning and serves as a starting point. We are not presenting this model as the answer; indeed more questions are raised by the success of such a simple implementation in predicting surface effects in this case study.

**L361: How is it that the essentially made up model (assuming a complete surface film) does better than the model that explicitly accounts for the system thermodynamics? Is this just saying that the thermodynamic model is wrong (e.g. bad parameterization or parameterization used outside the RH range where it is valid)? Or is there perhaps some kinetic effect that must be accounted for? The discussion that comes a few sentences later seems to suggest perhaps the former. But, some additional and more explicit discussion is necessary, rather than a statement (line 375) that this finding is "intriguing".**

This is indeed an interesting point, and the answer is we don't know. Clearly the more rigorous thermodynamic models miss something, such as how interfacial energies between liquid phases (not considered explicitly) might drive the system towards enrichment of the organic phase. The AIOMFAC-CLLPS (w/ org film) presented here is a limiting case in this treatment of the system and is an educated first guess at the solution. Kinetic effects may be in play, but typically these would make activation harder (dissolution kinetics, water uptake kinetics etc.).

**Minor Comments:**

**Fig. 1: It would be good to change from the red/green color scheme to one that is more friendly to color blind people.**

Done.